

**THE LAST GLACIAL TERMINATION ON THE EASTERN FLANK OF THE CENTRAL**
**PATAGONIAN ANDES (47°S)**
William I. Henríquez[1,2], Rodrigo Villa-Martínez[3], Isabel Vilanova[4], Ricardo De Pol-Holz[3], and
Patricio I. Moreno[2,*]
[1]Victoria University of Wellington, Wellington, New Zealand
[2]Instituto de Ecología y Biodiversidad, Departamento de Ciencias Ecológicas, Universidad
de Chile, Casilla 653, Santiago, Chile
[3]GAIA-Antártica, Universidad de Magallanes, Avda. Bulnes 01855, Punta Arenas,
Chile [4]Museo Argentino de Ciencias Naturales Bernardino Rivadavia, Avda. Angel Gallardo
470, Buenos Aires, Argentina.
[*]Corresponding author: pimoreno@uchile.cl



ABSTRACT
Few studies have examined in detail the sequence of events during the last glacial
termination (T1) in the core sector of the Patagonian Ice Sheet (PIS), the largest ice mass
in the southern hemisphere outside Antarctica. Here we report results from Lago Edita
(47°8'S, 72°25'W, 570 m.a.s.l.), a small closed-basin lake located in a valley overridden by
eastward-flowing Andean glaciers during the Last Glacial Maximum (LGM). Lago Edita
shows glaciolacustrine sedimentation until 19,400 yr BP and a mosaic of cold-resistant,
hygrophilous conifers and rainforest trees, along with alpine herbs between 11,000-
19,400 yr BP. Increases in arboreal pollen at 13,200 and 11,000 yr BP led to the
establishment of forests near Lago Edita between 9000-10,000 yr BP. Our data suggest
that the PIS retreated at least ~90 km from its LGM limit between ~19,400-21,000 yr BP
and that scattered, low-density populations of cold-resistant hygrophilous conifers,
rainforest trees, high Andean and steppe herbs thrived east of the Andes during the LGM
and T1, implying high precipitation and SWW intensity at 47°S. We interpret large-
magnitude increases in arboreal vegetation as treeline-rise episodes driven by warming
pulses at 13,200 and 11,000 yr BP coupled with a decline in SWW influence at ~11,000 yr
BP, judging from the disappearance of cold-resistant hygrophilous trees and herbs. We
propose that the PIS imposed a regional cooling signal along its eastern, downwind margin
through T1 that lasted until the separation of the North and South Patagonian icefields
along the Andes. We posit that the withdrawal of glacial and associated glaciolacustrine
environments through T1 provided a route for the dispersal of hygrophilous trees and
herbs from the eastern flank of the central Patagonian Andes, contributing to the
afforestation of the western Andean slopes and pacific coasts of central Patagonia during
T1.



INTRODUCTION

The Patagonian ice sheet (PIS) was the largest ice mass in the southern hemisphere
outside Antarctica during the last glacial maximum (LGM). Outlet lobes from the PIS
flowed westward into the Pacific coast south of 43ºS and eastward toward the extra-
Andean Patagonian plains, blanketing a broad range of environments and climatic zones
across and along the Andes. Land biota from formerly ice-free sectors underwent local
extinction or migrated toward the periphery of the advancing PIS during the last glaciation
until its culmination during the LGM. The PIS then underwent rapid recession and thinning
through the last glacial termination (termination 1= T1: between ~11,000-18,000 yr BP)
toward the Andes as illustrated by stratigraphic, geomorphic and radiocarbon-based
chronologies from northwestern Patagonia (39º-43ºS) (Denton et al., 1999; Moreno et al.,
2015). These data, along with the Canal de la Puntilla-Huelmo pollen record (Moreno et
al., 2015), indicate abandonment from the LGM margins in the lowlands at 17,800 yr BP
and accelerated retreat that exposed Andean cirques located above 800 m.a.s.l. within
1000 years or less in response to abrupt warming. Similarly, glaciers from Cordillera
Darwin (54º-55ºS), the southernmost icefield in South America, underwent rapid
recession from their LGM moraines located in central and northern Tierra del Fuego prior
to 17,500 yr BP, and led to ice-free conditions by 16,800  yr BP near the modern ice fronts
(Hall et al., 2013).
Because very few studies have been conducted in the continental sector of central-
west Patagonia (45º-48ºS) it is yet unclear (i) the timing of the LGM and the
structure/chronology of glacial retreat, (ii) the timing, structure and rates of climate
changes during T1, as well as the (iii) composition of the vegetation that thrived adjacent
to the LGM margins, (iv) the tempo and mode of vegetation colonization at site-specific
scale, and (v) at regional scale through the increasingly ice-free Patagonian landscapes
during T1. The latter is important for identifying possible glacial refugia and the dispersal
routes of the vegetation following the LGM.



Paleoclimate simulations (Bromwich et al., 2005; Bromwich et al., 2004) and
stratigraphic studies (Kaufman et al., 2004) in the periphery of the Laurentide Ice Sheet in
North America, have detected that large ice sheets exerted important impacts on the
thermal structure and atmospheric circulation at regional, continental and zonal scale
from the LGM to the early Holocene. This aspect has remained largely unexplored in the
PIS region, and might be a factor of importance for understanding the dynamics of the
SWW and climatic/biogeographic heterogeneities through T1 at regional scale. Progress in
this field requires understanding the deglacial chronology of the PIS and a suite of
sensitive paleoclimate sites across and along the residual ice masses through the last
transition from extreme glacial to extreme interglacial conditions.
Recent chronologies based on cosmogenic radio nuclides of terminal moraines of
the Río Blanco and recessional moraines deposited by the Lago Cochrane ice lobe (LCIL)
(Boex et al., 2013; Hein et al., 2010) (Figure 1), and optically stimulated luminescence
dating of glaciolacustrine beds associated with Glacial Lake Cochrane (GLC) (47ºS) (Glasser
et al., 2016) reported ages of 29,000 yr BP for the final LGM advance and an interval
between 8000-13,000 yr BP for the subsequent drainage of GLC toward the Pacific, event
that took place when enough glacial recession and thinning breached the continuity that
the North and South Patagonian Icefields achieved during the LGM (Turner et al., 2005).
Palynological interpretations from the Lago Shaman and Mallín Pollux sites (de Porras et
al., 2012; Markgraf et al., 2007), located east of the Andes between 44ºS and 45ºS
respectively (Figure 1), indicate predominance of cold and dry conditions during T1 and
negative anomalies in southern westerly wind (SWW) influence. The validity and regional
applicability of these stratigraphic, chronologic and palynologic interpretations, however,
awaits replication by detailed stratigraphic/geomorphic data from sensitive sites
constrained by precise chronologies.
In this study we report high-resolution pollen and macroscopic charcoal records
from sediment cores we collected in Lago Edita (47°8'S, 72°25'W, ~570 m.a.s.l.), a small
closed-basin lake located in Valle Chacabuco, east of the central Patagonian Andes (Figure
1). Stratigraphic and chronologic results from Valle Chacabuco are important for



elucidating the timing and rates of deglaciation in this core region of the PIS because this
valley is located approximately two thirds (90 km) upstream from the LGM moraines
deposited by LCIL east of Lago Cochrane relative to the modern ice fronts, and its
elevation spans the highest levels of GLC during T1. The Lago Edita data allow assessment
of vegetation, fire-regime and climate changes during the last global transition from
extreme glacial to extreme interglacial conditions in central-west Patagonia. The aim of
this paper is to contribute toward: (1) the development of a recessional chronology of the
LCIL and (2) regressive phases of GLC, (3) documenting the composition and geographic
shifts of the glacial and deglacial vegetation, (4) understanding the tempo and mode of
vegetation and climate changes during T1 and the early Holocene, (5) constraining the
regional climatic influence of the PIS during T1 in terrestrial environments, and (6)
identifying possible dispersal routes of tree taxa characteristic of modern evergreen
forests in central-west Patagonia during T1.

Study Area

Central Chilean Patagonia, i.e. the Aysén region (43°45'S-47°45'S), includes

numerous channels, fjords, islands, and archipelagos along the Pacific side, attesting for
tectonic subsidence of Cordillera de la Costa and intense glacial erosion during the
Quaternary. The central sector features an intricate relief associated to the Patagonian
Andes with summits surpassing 3000 m.a.s.l., deep valleys, lakes of glacial origin, and
active volcanoes such as Hudson, Macá, Cay, Mentolat and Melimoyu (Stern, 2004). The
Andes harbors numerous glaciers and the North Patagonian Icefield (Figure 1), which
acted as the source for multiple outlet glacier lobes that coalesced with glaciers from the
South Patagonian Icefield and formed the PIS during Quaternary glaciations, blocked the
drainage toward the Pacific and changed the continental divide in the region (Turner et
al., 2005). Farther to the east the landscape transitions into the back-arc extra-Andean
plains and plateaus.



Patagonia is ideal for studying the paleoclimate evolution of the southern mid-
latitudes including past changes in the SWW because it is the sole continental landmass
that intersects the low and mid-elevation zonal atmospheric flow south of 47°S.
Orographic rains associated to storms embedded in the SWW enhance local precipitation
by the ascent of moisture-laden air masses along the western Andean slopes, giving way
to subsidence and acceleration of moisture-deprived winds along the eastern Andean
slopes (Garreaud et al., 2013). This process accounts for a steep precipitation gradient
across the Andes, illustrated by the annual precipitation measured in the coastal township
of Puerto Aysén (2414 mm/year) and the inland Balmaceda (555 mm/year)
(http://explorador.cr2.cl/), localities separated by ~80 km along a west-to-east axis. The
town of Cochrane, located ~15 km south of our study site features annual precipitation of
680 mm/year and mean annual temperature of 7.8 °C (Figure 1).
Weather station and reanalysis data along western Patagonia show positive
correlations between zonal wind speed and local precipitation, a relationship that extends
to sectors adjacent to the eastern slopes of the Andes (Garreaud et al., 2013; Moreno et
al., 2014). Therefore, changes in local precipitation in the Aysén region are good
diagnostics for atmospheric circulation changes associated with the frequency/intensity of
storms embedded in the SWW over a large portion of the southeast Pacific. This
relationship can be applied to paleoclimate records from central Chilean Patagonia for
inferring the behavior of the SWW on the basis of past changes in precipitation or
hydrologic balance.
The steep precipitation gradient, in conjunction with adiabatic cooling and enhanced
continentality toward the east, influences the distribution and composition of the
vegetation, inducing altitudinal, latitudinal and longitudinal zonation of plant communities
throughout the Patagonian Andes. Physiognomic and floristic studies (Gajardo, 1994;
Luebert and Pliscoff, 2006; Pisano, 1997; Schmithüsen, 1956) have recognized five units or
communities which we characterize succinctly in the following sentences: 1) Magellanic
Moorland: this unit occurs in maritime sectors with high precipitation (3000-4000
mm/year and low seasonality) along the islands, fjords and channels, it is dominated by



cushion-forming plants such as *Donatia fascicularis*, *Astelia pumila* and *Tetroncium*
*magallanicum*. Also present are the hygrophilous cold-resistant trees *Nothofagus*
*betuloides* and the conifers *Pilgerodendron uviferum*, *Lepidothamnus fonkii* and
*Podocarpus nubigena*. 2) Evergreen rainforest: present in humid, temperate (1500 -3000
mm/year; <600 m.a.s.l.) sectors of Aysén, this unit is characterized by the trees
*Nothofagus nitida*, *N. betuloides*, *Drimys winteri*, along with *P. uviferum* in waterlogged
environments. 3) Winter deciduous forests: located in relatively cooler and/or drier
sectors with higher seasonality (400-1000 mm/year; 500-1250 m.a.s.l.). The dominant tree
is *Nothofagus pumilio*, which intermingles with *N. betuloides* in western sites and the
Patagonian steppe eastward. In the latter *N. pumilio* forms monospecific stands and
presents a species-poor understory. 4) Patagonian steppe: occurs in substantially drier
(<500 mm/year) lowland areas with heightened continentality. This unit is dominated by
herbs of the families Poaceae (*Festuca, Deschampsia, Stipa, Hordeum, Rytidosperma,*
*Bromus, Elymus*), Rubiaceae (*Galium*), and shrubs of families Apiaceae (*Mulinum*),
Rosaceae (*Acaena*), Fabaceae (*Adesmia*) and Rhamnaceae (*Discaria*). 5) High Andean
Desert: occurs in the wind-swept montane environments above the treeline (>1000
m.a.s.l.) and is represented by herbs of the families Poaceae (*Poa, Festuca*), Asteraceae
(*Nassauvia*, *Senecio*, *Perezia*), Berberidaceae (*Berberis*), Brassicaceae (*Cardamine*),
Santalaceae (*Nanodea*), Rubiaceae (*Oreopulus*) Apiaceae (*Bolax*), Ericaceae (*Gaultheria,*
*Empetrum*), along with *Gunnera magellanica* and *Valeriana*, with occasional patches of
*Nothofagus antarctica*.

MATERIALS AND METHODS

We collected overlapping sediment cores over the deepest sector of Lago Edita (8 m

water depth) from an anchored coring rig equipped with 10-cm diameter aluminum casing
tube, using a 5-cm diameter Wright piston corer and a 7.5-cm diameter sediment-water
interface piston corer with a transparent plastic chamber. We characterized the
stratigraphy through visual descriptions, digital X radiographs to identify stratigraphic



structures and loss-on-ignition to quantify the amount of organic, carbonate and
siliciclastic components in the sediments (Heiri et al., 2001).
The chronology of the record is constrained by AMS radiocarbon dates on bulk
sediment and chronostratigraphic correlation of the H1 tephra from Volcán Hudson (Stern
et al., 2016). The radiocarbon dates were calibrated to calendar years before present (yr
BP) using the CALIB 7.0 program. We developed a Bayesian age model using the Bacon
package for R (Blaauw and Christen, 2011) to assign interpolated ages and confidence
intervals for each level analyzed.
We processed and analyzed continuous/contiguous sediment samples (2 cc) for
pollen and fossil charcoal. The samples were processed using a standard procedure that
includes 10% KOH, sieving with a 120 μm mesh, 46% HF and acetolysis (Faegri and Iversen,
1989). We added exotic *Lycopodium* spores tablets to calculate concentration
(particles*cc) and accumulation rates of pollen and microscopic charcoal (particles*cm$^{-2}$*years$^{-1}$) from each level. We counted between 200-300 pollen grains produced by trees,
shrubs and herbs (terrestrial pollen) for each palynological sample and calculated the
percent abundance of each terrestrial taxon relative to this sum. The percentage of
aquatic plants was calculated in reference to the total pollen sum (terrestrial + aquatic
pollen) and the percentage of ferns from the total pollen and spores sum. Zonation of the
pollen record was aided by a stratigraphically constrained cluster analysis on all terrestrial
pollen taxa having ≥2%, after recalculating sums and percentages.
We identified the palynomorphs based on a modern reference collection housed at
the laboratory of Quaternary Paleoecology of Universidad de Chile, along with published
descriptions and keys (Heusser, 1971). In most cases the identification was done at family
or genus level, in some cases to the species level (*Podocarpus nubigena*, *Drimys winteri*,
*Gunnera magellanica*, *Lycopodium magellanicum*). The palynomorph *Nothofagus dombeyi*
type includes the species *N. antarctica*, *N. pumilio*, *N. betuloides* and *N. dombeyi*, the
morphotype *Fitzroya/Pilgerodendron* includes the cupressaceous conifers *Fitzroya*
*cupressoides* and *Pilgerodendron uviferum*.



We tallied microscopic (<120 µm) and macroscopic (>106 µm) charcoal particles to
document regional and local fire events, respectively. Microscopic particles were counted
from each pollen slide, while macroscopic charcoal was counted from 2-cc sediment
samples obtained from 1-cm thick and continuous-contiguous sections. The samples were
prepared using a standard procedure which involves deffloculation in 10% KOH, careful
sieving through 106 and 212 µm-diameter meshes to avoid rupture of individual particles,
followed by visual inspection on a ZEISS KL 1500 LCD stereoscope at 10x magnification.
These results were analyzed by a time-series analysis to detect local fire events using the
CharAnalysis software (Higuera et al., 2009), interpolating samples at regular time interval
based in the median time resolution of the record. We deconvoluted the CHAR signal into
a peaks and background component using a lowess robust to outliers smoothing with a
100-yr window width. We calculated locally defined thresholds to identify statistically
significant charcoal peaks or local fires events (99[th] percentile of a Gaussian distribution).

RESULTS

The sediment stratigraphy (Figure 2) reveals a basal unit of blue-gray mud between
819-1726 cm, horizontally laminated for the most part, in some sectors massive and
sandier with small amounts of granule and gravel immersed in a clayey matrix (segment
PC0902AT9). These inorganic clays are overlain by organic silt between 678-819 cm and
organic-rich lake mud (gytjja) in the topmost 678 cm. We found laminated carbonates
between 759-794 and 389-394 cm, for the remainder of the record carbonate values are
negligible or null. The record includes 2 tephras between 628-630 and 643-661 cm, which
exhibit sharp horizontal contacts with the over and underlying mud and, consequently, we
interpret them as aerial fallout deposits from explosive events originated from Volcán
Hudson (H1 tephra) and from Volcán Mentolat (M1 tephra) based on geochemical data,
respectively (Stern et al., 2016).
The radiocarbon results show an approximately linear increase of age with depth
between 9000-19,000 yr BP (Figure 3) which, in conjunction with the sediment



stratigraphy, suggests undisturbed in-situ pelagic deposition of lake mud and tephras in
the Lago Edita basin. This study focuses on the interval between 9000-19,000 yr BP (Figure
2, Table 1), and consists of 155 contiguous palynological and macroscopic charcoal levels
with a median time step of 65 years between analyzed samples.

Pollen stratigraphy

We divided the record in 6 zones to facilitate its description and discussion, based
on conspicuous changes in the pollen stratigraphy and a stratigraphically constrained
cluster analysis (Figure 4). The following section describes each pollen zone indicating the
stratigraphic and chronologic range, and the mean abundance of major taxa in
parenthesis.
Zone Edita-1 (780-795 cm; 18,100-19,000 yr BP) is co-dominated by Poaceae (33%)
and *Empetrum* (32%). This zone starts with a gradual increase in *Empetrum*, attaining its
maximum abundance (~53%) at the end of this zone. Asteraceae subfamily Asteroideae
(7%), *Acaena* (4%), Caryophyllaceae (3%) and Cyperaceae (9%) decrease, while Poaceae
shows fluctuations in its abundance between 2-16 % over the entire interval. Other herbs
and shrubs such as Ericaceae (3%), *Phacelia* (~2%), *Valeriana* (1%), *Gunnera magellanica*
(~2%), Apiaceae (<1%), and Asteraceae subfamily Cichorioideae (<1%) remain relatively
steady. The arboreal taxa *N. dombeyi* type (10%), *Fitzroya/Pilgerodendron* (2%), *P.*
*nubigena* (<1%) and *D. winteri* (<1%) are present in low abundance, as well as the ferns *L.*
*magellanicum* (~1%) and *Blechnum* type (5%) and the green-microalgae *Pediastrum* (2%).
Zone Edita-2 (758-780 cm; 16,800-18,100 yr BP) begins with a decline in *Empetrum*
(30%) and an increase in Poaceae (34%) followed by its decrease until the end of this zone.
*N. dombeyi* type (15%), Caryophyllaceae (5%) and Asteraceae subfamily Asteroideae (5%)
show a rising trend during this zone, while other arboreal taxa (*Fitzroya/Pilgerodendron*
(3%), *P. nubigena* (<1%) and *D. winteri* (<1%) and most of the herbs maintain similar
abundance the previous zone. *L. magellanicum* (2%) and *Pediastrum* (4%) rise slightly,
along with high variability in Cyperaceae (7%).
Zone Edita-3 (701-758 cm; 13,200-16,800 yr BP) is characterized by a sharp rise in
Poaceae (45%) and declining trend in *Empetrum* (15%). The conifer *P. nubigena* (2%) starts
a sustained increase, while *N. dombeyi* type (13%) and *Fitzroya/Pilgerodendron* (3%)
remain relatively invariant. *D. winteri* (<1%) and *Misodendrum* (<1%), a mistletoe that
grows on *Nothofagus* species, appear in low abundance in an intermittent manner.
*Pediastrum* (30%) shows a rapid increase until 15,600 yr BP, followed by considerable
variations in its abundance until the end of this zone (between 19% and 55%). *L.*
*magellanicum* (3%) shows a steady increase, while *Blechnum* type (6%) remains invariant
and Cyperaceae (7%) exhibits large fluctuations superimposed upon a declining trend.
Zone Edita-4 (681-701 cm; 11,600-13,200 yr BP) starts with step increases in *N.*
*dombeyi* type (29%) and *Misodendrum* (1%). *P. nubigena* (5%) starts this zone with
variability and stabilizes toward the end of this zone, concurrent with
*Fitzroya/Pilgerodendron* (3%) and traces of *D. winteri* (<1%). Poaceae (38%) shows a
steady decrease, while *Empetrum* (6%) continues with a declining trend that started
during the previous zone. Asteraceae subfamily Asteroideae (5%) and Caryophyllaceae
(2%) decrease, *L. magellanicum* (3%), Cyperaceae (4%) and *Pediastrum* (24%) decline
gradually with considerable fluctuations, while *Blechnum*- type (11%) shows modest
increases.
Zone Edita-5 (674-681 cm; 11,100-11,600 yr BP) shows a marked decline in *N.*
*dombeyi* type (27%), *Misodendrum* (<1%) and Poaceae (33%) in concert with a
conspicuous increase in the conifers *Fitzroya/Pilgerodendron* (12%) and *P. nubigena* (9%)
that reach their peak abundance in the record. The abundance of herbs and shrubs
decreases or remains steady, with the exception of an ephemeral increase in *Phacelia*
(3%). *Blechnum* type (39%) shows a remarkable increase to its peak abundance in the
entire record, while *L. magellanicum* (3%), Cyperaceae (8%) and *Pediastrum* (17%) rise
slightly.
Zone Edita-6 (640-674 cm; 8940-11,100 yr BP) is characterized by an abrupt increase
in *N. dombeyi* type (62%) and *Misodendrum* (2%), along with conspicuous decline in
*Fitzroya/Pilgerodendron* (2%) and *P. nubigena* (2%) at the beginning of this zone. Poaceae





(26%) shows a downward trend over this period, while others herbs and shrubs
(*Empetrum*, Ericaceae, Caryophyllaceae, Asteraceae subfamily Asteroideae, *Acaena*,
*Phacelia*, *Valeriana*, *Gunnera magellanica*, Apiaceae and Asteraceae subf. Cichorioideae)
show their lowest abundance in the record. *Blechnum* type (7%) drops sharply, followed
by a gradual decline in concert with *L. magellanicum* (1%). Cyperaceae (7%) and
*Pediastrum* (6%) show initial declines followed by increases toward the end of this zone.

Charcoal stratigraphy

The record from Lago Edita shows absence of macroscopic charcoal particles

between 14,300-19,000 yr BP followed by an increase in charcoal accumulation rate
(CHAR) that led to a variable plateau between 12,000-13,200 yr BP, a 1000-year long
decline, and a sustained increase led to peak abundance at 9700 yr BP. Charcoal values
then declined rapidly to intermediate levels by 9000 yr BP. We note a close
correspondence between the *Nothofagus* abundance (%) and the CHAR suggesting that
charcoal production was highly dependent upon quantity and spatial continuity of coarse
woody fuels in the landscape (Figure 5).

Time-series analysis of the macroscopic charcoal record revealed 11 statistically

significant peaks we interpret as local fires events within the Lago Edita watershed (Figure
5). The temporal structure of these events indicates a sequence of millennial-scale peaks
in fire frequency with maxima at 9600, 10,900, 12,000, 13,100, and 14,100 yr BP. We
observe a steady increase in the fire frequency maxima from 14,100 to 10,900 yr BP
(Figure 5).

DISCUSSION
Paleovegetation

The pollen record from Lago Edita (Figures 4, 6) documents dominance of herbs and

shrubs (chiefly Poaceae, *Empetrum*, Asteraceae, accompanied by Caryophyllaceae,



*Acaena*, Ericaceae, *Phacelia*, *Valeriana*, and Apiaceae in lower abundance) found above
the modern treeline and the Patagonian steppe between 11,000-19,000 yr BP, followed by
increasing *Nothofagus* we interpret as the establishment of scrubland (11,000-13,200 yr
BP), woodland (10,500-11,000 yr BP) and forest (9000-10,500 yr BP). Within the interval
dominated by non-arboreal taxa we distinguish an initial phase with abundant *Empetrum*
between 16,800-19,000 yr BP, followed by diversification of the herbaceous assemblage
and preeminence of Poaceae during the interval 11,000-16,800 yr BP (Figures 4, 6). This
change is contemporaneous with a sustained rise of *P. nubigena* and the mistletoe
*Misodendrum* coeval with conspicuous increases in *Lycopodium magellanicum* and the
green microalga *Pediastrum.* We emphasize the continuous presence of the arboreal
*Nothofagus* and *Fitzroya/Pilgerodendron* in low but constant abundance (~15% and ~3%,
respectively) between 13,000-19,000 yr BP, along with traces (<3%) of hygrophilous trees
(*Podocarpus nubigena, Drimys winteri*) and herbs (*Gunnera magellanica, Lycopodium*
*magellanicum*) accounting, in sum, for a persistent ~25% of the pre-13,200 yr BP pollen
record (Figures 4, 6).

The mixed palynological assemblage between ~11,000-19,400 yr BP has no modern

analogues in the regional vegetation (Luebert and Pliscoff, 2006; Mancini, 2002). Possible
explanations for its development involve: (a) downslope migration of High Andean
vegetation driven by snowline and treeline lowering associated with intense glaciation in
the region, coupled with (b) the occurrence of scattered, low-density populations of
hygrophilous trees and herbs along the eastern margin of the PIS during the LGM and T1.
We rule out the alternative explanation that pollen grains and spores of hygrophilous
trees and herbs in Lago Edita represent an advected signal through the Andes from ice-
free humid Pacific sectors harboring these species because: (i) no empirical basis is
currently available for ice-free conditions and occurrence of cold-resistant hygrophilous
taxa along the western Andean slopes or the Pacific coast of central Patagonia during the
LGM; in fact, the oldest minimum limiting dates for ice-free conditions in records from
Taitao Peninsula and the Chonos archipelago yielded ages of 14,335±140 and
13,560±125 $^{14}$C yr BP (median age probability [MAP]: 17,458 and 16,345 yr BP),



respectively (Haberle and Bennett, 2004; Lumley and Switsur, 1993); (ii) the appearance of
*Fitzroya/Pilgerodendron* and *Podocarpus nubigena* at ~15,000 and ~14,000 yr BP,
respectively, occurred 4000-5000 years later in coastal Pacific sites relative to the Lago
Edita record (Figure 7); (iii) background levels of *Nothofagus* between 15-20% in Lago
Edita predate the appearance and expansion of this taxon in coastal Pacific sites and, once
realized, its abundance in Lago Edita did not follow the trend and magnitude observed in
western sites, as expected if the palynological signal in Lago Edita was attributed to long-
distance transport from that source (Figure 7).

Previous palynological studies from bogs located east of the central Patagonian

Andes (de Porras et al., 2012; Markgraf et al., 2007) interpreted dry conditions prior to
~12,000 yr BP, based on the premise that low abundance of arboreal taxa and
preeminence of herbs and shrubs were indicative of Patagonian Steppe communities. The
glacial-to-interglacial vegetation change in those studies was interpreted as a westward
shift of the forest-steppe boundary brought by lower-than-present SWW influence at 44°-
46°S, followed by a rise in temperature and precipitation at the end of the last glaciation.
In contrast, the Lago Augusta site (located in Valle Chacabuco ~7 km northeast of Lago
Edita) (Figure 1) shows a pollen assemblage prior to 15,600 yr BP dominated by high
Andean herbs and shrubs, along with taxa characteristic of hyperhumid environments
along the Pacific coasts of central Patagonia (*Nothofagus, Fitzroya/Pilgerodendron,*
*Podocarpus nubigena, Saxegothaea conspicua, Drimys winteri, Dysopsis glechomoides* and
the ferns *Blechnum*, Hymenophyllaceae, *Cystopteris*) (Villa-Martinez et al., 2012). It
appears then that floristic elements of modern Patagonian forests were present in low
abundance and in a discontinuous manner along the eastern flank of the PIS between 44°-
47°S. The data shown in this paper, along with the results from Lago Augusta, suggest that
Valle Chacabuco harbored cryptic refugia (Bennett and Provan, 2008) of rainforest trees
and herbs during the interval 11,000-19,000 yr BP, therefore the interpretation of lower-
than-present precipitation of SWW origin in previous studies (de Porras et al., 2012;
Markgraf et al., 2007), is not applicable to the Valle Chacabuco area over this time
interval. Plant colonization of Valle Chacabuco must have started from the LGM limits



located east of Lago Cochrane and followed the shrinking ice masses to the west, once the
newly deglaciated sectors were devoid of glaciolacustrine influence through T1.
Declines and virtual disappearance of the cold-resistant hygrophilous trees
*Fitzroya/Pilgerodendron*, *Podocarpus nubigena* and the herbs *Gunnera magellanica* and
*Lycopodium magellanicum* took place at ~11,000 yr BP in the Lago Edita record (Figures 4,
6), in response to a sudden decline in precipitation. These changes were
contemporaneous with a sustained rise in *Nothofagus*, decreases in all other shrubs and
herbs, and a major increase in macroscopic charcoal (Figure 5), signaling an increment in
arboreal cover, higher spatial continuity of coarse fuels and forest fires. We interpret this
arboreal increase and fire-regime shift as driven by warming which might have triggered a
treeline rise and favored the spread/densification of woody species and coarse fuels
(Figures 4, 5, 6). *Nothofagus* forests (~70% abundance) established near Lago Edita
between 9000-10,000 yr BP.

Deglaciation of Valle Chacabuco and the Lago Cochrane basin

Stratigraphic and chronologic results from Lago Edita are key for deciphering the
evolution of Valle Chacabuco and for constraining the timing and rates of deglaciation in
this core region of the PIS. Previous studies (Hein et al., 2010) indicate that Valle
Chacabuco was overridden by the Lago Cochrane ice lobe (LCIL) during the LGM and
deposited the Río Blanco moraines ~90 km downstream from Lago Edita, distal to the
eastern end of Lago Cochrane in Argentina (Argentinian name: Lago Pueyrredón).
Cosmogenic radionuclide dating on the Río Blanco moraine belts yielded ages of
19,100±700, 22,800±1000 and 26,000±900 yr BP (Hein et al., 2010). Kaplan et al. (2011)
recalculated these ages using a local production rate constrained by radiocarbon dates
from southern Patagonia and produced ages of ~21,100, ~25,100, and ~28,700 yr BP
respectively. This was followed by glacial recession starting at 17,400±700 (recalculated
age: 19,600±800) yr BP, formation of Glacial Lake Cochrane (GLC), stabilization and
deposition of the Lago Columna and Lago Posada moraines at 15,900±800 (recalculated



age: 17,600±900) yr BP, ~55 km upstream from the Río Blanco moraines (Hein et al., 2010;
Kaplan et al., 2011) (Figure 1). Further glacial recession led to the westward expansion and
lowering of GLC until the LCIL stabilized and deposited moraines in Lago Esmeralda
between 12,800-13,600 yr BP ~60 km upstream from the Lago Columna and Lago Posada
moraines (Turner et al., 2005). Recession from this position led to sudden drainage of GLC
toward the Pacific Ocean via Río Baker, once the continuity between the North and South
Patagonian icefields was breached by glacial recession and thinning. According to these
data Valle Chacabuco may have been ice-free and devoid of glaciolacustrine influence
after ~17,600 yr BP. More recently Boex et al. (2013) reported a cosmogenic radio nuclide-
based reconstruction of vertical profile changes of the LCIL through the LGM and T1 that
reveals deposition of (i) the Sierra Colorado lower limit by 28,980±1206 yr BP which can
be traced to the Río Blanco moraines, (ii) the highest summits of Cerro Oportus and Lago
Columna moraines by 18,966±1917 yr BP, and (iii) the María Elena moraine by
17,088±1542 yr BP. According to these data Valle Chacabuco may have been ice-free after
~17,000 yr BP.

Lago Edita is a closed-basin lake located ~11 km east of the Cerro Tamango summit

along the ridge that defines the southern edge of the Valle Chacabuco watershed (Figure
1). Lacustrine sedimentation in Lago Edita started when ice-free conditions developed in
Valle Chacabuco, as the LCIL snout retreated eastward to a yet unknown position. The
Lago Edita cores show 9 meters of blue-gray clays with millimeter-scale laminations,
interrupted by sporadic intervals of massive pebbly mud appreciable in x radiographs and
the LOI$_{550}$ record as increases in the inorganic density data (Figure 2).  We also found
exposed glaciolacustrine beds and discontinuous fragments of lake terraces in the vicinity
of Lago Edita, attesting for a large lake that flooded Valle Chacabuco in its entirety.
Differential GPS measurements of 570 m.a.s.l. for the Lago Edita surface and 591 m.a.s.l.
for a well-preserved terrace fragment located ~150 m directly south of Lago Edita, provide
minimum-elevation constraints for GLC during this stage. The Lago Augusta site (Villa-
Martinez et al., 2012), located ~7 km northeast of Lago Edita on the Valle Chacabuco floor



at 444 m.a.s.l. (Figure 1), shows 8 meters of basal glaciolacustrine mud (Figure 2) lending
support to our interpretation.
Glaciolacustrine sedimentation persisted in Lago Edita and Lago Augusta until the
surface elevation of GLC dropped below 570 and 444 m.a.s.l., respectively, and the closed-
basin lakes developed. The chronology for this event is constrained by statistically
identical AMS dates of 16,250±90 and 16,020±50 [14]C yr BP (UCIAMS-133418 and CAMS-
144454, respectively) (Table 1) from the same level in the basal portion of the organic
sediments in the Lago Edita record; this estimate approaches the timing for the cessation
of glaciolacustrine influence in Lago Augusta, radiocarbon-dated at 16,445±45 [14]C yr BP
(CAMS-144600) (Table 1). Because we observe approximately the same age for the
transition from glaciolacustrine to organic-rich mud in both stratigraphies, we interpret
the weighted mean age of those three dates (16,254±63 [14]C yr BP, MAP: 19,426 yr BP, two
different laboratories) as a minimum-limiting age for ice-free conditions and nearly
synchronous glaciolacustrine regression from elevations 591 and 444 m.a.s.l. in Valle
Chacabuco. We acknowledge that Villa-Martínez et al. (2012) excluded the age of date
CAMS-144600 from the age model of the Lago Augusta palynological record because it
was anomalously old in the context of other radiocarbon dates higher up in core.
Comparison of the radiocarbon-dated stratigraphy from Lago Edita record with the
exposure-age-dated glacial geomorphology from Lago Cochrane/Pueyrredón, Valle
Chacabuco and surrounding mountains reveals the following:
• The geochronology for the innermost (third) belt of Río Blanco moraines (~21,100
yr BP) (Hein et al., 2010; Kaplan et al., 2011), glacial deposits on the highest
summits of Cerro Oportus and the Lago Columna moraines (18,966±1917 yr BP)
(Boex et al., 2013) are compatible (within error) with the onset of organic
sedimentation in Lago Edita and Lago Augusta at 19,426 yr BP in Valle Chacabuco.
If correct, this implies ~90 km recession of the LCIL from its LGM limit within ~1500
years.
• Hein et al. (2010)′s chronology for the "final LGM limit", Lago Columna and Lago
Posada moraines are anomalously young, as well as Boex et al. (2013)'s chronology



for the María Elena moraine. This is because cosmogenic radio nuclide ages for
these landforms postdate the onset of organic sedimentation in Lago Edita and
Lago Augusta, despite being morphostratigraphically distal (older) than Valle
Chacabuco.
• As shown in Figure 1, Lago Edita is located along a saddle that establishes the
southern limit of the Río Chacabuco catchment and the northern limit of the Lago
Cochrane basin. According to Hein et al. (2010) the drainage divide on the eastern
end of Lago Cochrane/Pueyrredón basin is located at 475 m.a.s.l., therefore the
presence of this perched glacial lake with a surface elevation of 591 m.a.s.l.
requires ice dams located in the Valle Chacabuco and the Lago Cochrane basin.
This suggests that both valleys remained partially ice covered and that enough
glacier thinning and recession early during T1 enabled the development of a
topographicaly constrained glacial lake that covered Valle Chacabuco up to the
aforementioned saddle.
• The high stand of GLC at 591 m.a.s.l. lasted for less than 1500 years during the
LGM and was followed by a nearly instantaneous lake-level lowering of at least
~150 m at ~19,400 yr BP in Valle Chacabuco. The abrupt large-magnitude drainage
event of this "predecessor lake" was recently recognized by Bourgois et al. (2016),
but its chronology, hydrographic and climatic implications have not been
addressed in the literature.

Biogeographic and paleoclimatic implications

The persistence of scattered, low-density populations of rainforest trees and herbs

east of the Andes during the LGM and T1 (Figures 4, 6) implies that precipitation delivered
by the SWW must have been substantially higher than at present (680 mm/year measured
in the Cochrane meteorological station). Because local precipitation in western Patagonian
is positively and significantly correlated with low-level zonal winds (Garreaud et al., 2013;
Moreno et al., 2010; Moreno et al., 2014), we propose that the SWW influence at 47°S



was stronger than present between 11,000-19,000 yr BP, in particular between 11,000-
16,800 yr BP. Subsequent increases in arboreal vegetation, chiefly *Nothofagus,* at 11,000
and 13,200 yr BP led to the establishment of forests near Lago Edita between 9000-10,000
yr BP (Figures 4, 6). We interpret these increases as treeline-rise episodes driven by
warming pulses coupled with a decline in SWW strength at 47°S, as suggested by the
disappearance of cold-resistant hygrophilous trees and herbs at ~11,000 yr BP. We
speculate that the warm pulse and decline in SWW influence at ~11,000 yr BP might
account for the abandonment of early Holocene glacier margins in multiple valleys in
central Patagonia (Glasser et al., 2012)

Four salient aspects of the Lago Edita record are relevant for deciphering the pattern

and rates of climate change and dispersal routes of the vegetation in Central Patagonia
(47°S) during the last glacial termination (T1):

1-  Absence of stratigraphically discernable indications of deglacial warming

between 13,200-19,400 yr BP, in contrast to northwestern Patagonian records

(the Canal de la Puntilla and Huelmo sites) (Moreno et al., 2015) which show that

75-80% of the glacial-interglacial temperature recovery was accomplished

between 16,800-17,800 yr BP (Figure 8). The record from Lago Stibnite, located

in central-west Patagonia upwind from the PIS and Lago Edita, shows a rapid

increase in arboreal pollen from ~2% to >80% in less than 1000 years starting at

16,200 yr BP (Figure 8). We posit that cold glacial conditions lingered along the

periphery of the shrinking PIS during T1, affecting adjacent downwind sectors

such as Valle Chacabuco. According to Turner et al. (2005) the LCIL stabilized and

deposited moraines in Lago Esmeralda, located ~10 km upstream and ~240 m

lower in elevation than Lago Edita, between 12,800-13,600 yr BP. We propose

that the climatic barrier for arboreal expansion vanished in downwind sectors

such as Valle Chacabuco once glacial recession from the Lago Esmeralda margin

breached the continuity of the North and South Patagonian icefields along the

Andes. Thus, we propose that regional cooling induced by the PIS along its



eastern margin through T1 accounts for the delayed warming in Valle Chacabuco
relative to records located in western and northwestern sectors (Figure 8).
2- Cold and wet conditions prevailed between 16,800-19,400 yr BP, followed by an
increase in precipitation at 16,800 yr BP. The latter event is contemporaneous
with the onset of a lake-level rise in Lago Lepué (43°S, central-east Isla Grande
de Chiloé) (Figure 8), which Pesce & Moreno (2014) interpreted as a northward
shift of the SWW as they recovered from a prominent southward shift from
latitudes ~41°-43°S (Figure 8) following the onset of T1 (Moreno et al., 2015).
3- Significant ice recession (~90 km) from the eastern LGM margin of the Lago
Cochrane Ice lobe (LCIL) was accomplished between ~19,400-21,000 yr BP, at
times when northwestern Patagonian piedmont glacier lobes experienced
moderate recession during the Varas interstade (Denton et al., 1999; Moreno et
al., 2015) (Figure 8). In contrast to the LCIL, northwestern Patagonian piedmont
glacier lobes readvanced to their youngest glacial maximum position during a
cold episode between 17,800-19,300 yr BP that featured stronger SWW
influence at 41°-43°S (Moreno et al., 2015) (Figure 8). One explanation for this
latitudinal difference might be that northward-shifted SWW between 17,800-
19,300 yr BP fueled glacier growth in northwestern Patagonia while reducing the
delivery of moisture to central Patagonia, causing the LCIL to continue the
recession it had started during the Varas interstade.
4- A mosaic of cold-resistant and hygrophilous trees and herbs, currently found
along the humid western slopes of the Andes of central Chilean Patagonia, and
cold-resistant shrubs and herbs common to high-Andean and Patagonian steppe
communities developed along the eastern margin of the PIS during the LGM and
T1 (Figures 4, 6). We posit that glacial withdrawal and drainage of GLC through
T1 provided a route for the westward dispersal of hygrophilous trees and herbs,
contributing to the forestation of the newly deglaciated sectors of central-west
Patagonia.





We conclude that warm pulses at 13,200 and 11,000 yr BP and a decline in SWW
influence at 47°S starting at 11,000 yr BP brought T1 to an end in central-west Patagonia.
The earliest of these events overlaps in timing with the culmination of Patagonian (Garcia
et al., 2012; Moreno et al., 2009; Strelin et al., 2011; Strelin and Malagnino, 2000) and
New Zealand glacier advances during the Antarctic Cold Reversal. Our data suggest that
the subsequent warm pulse, which was accompanied by a decline in SWW strength at
11,000 yr BP (Moreno et al., 2010; Moreno et al., 2012), was the decisive event that led to
the end of T1 in the study area.

ACKNOWLEDGEMENTS
This study was funded by Fondecyt #1080485, 1121141, ICM grants P05-002 and
NC120066, and a CONICYT M.Sc. Scholarship to W.I.H. We thank E.A. Sagredo, O.H. Pesce,
E. Simi, and I. Jara for assistance during field work, K.D. Bennett and S. Haberle for sharing
published palynological data. We thank C. Saucedo from Agencia de Conservación
Patagónica for permission to work and collect samples in Hacienda Valle Chacabuco
(Parque Patagonia).




FIGURE AND TABLE CAPTIONS
Table 1. Radiocarbon dates from the Lago Edita core. The radiocarbon dates were
calibrated to calendar years before present using the CALIB 7.0 program.

Figure 1. Sketch map of the study area showing the location of central-west Patagonia, the
position of Valle Chacabuco relative to the Río Blanco ice limit east Lago of Cochrane, and
the North Patagonian icefield and Peninsula Taitao to the west. The lower portion of the
figure shows a detail on the Cerro Tamango area and the portion of Valle Chacabuco
where Lago Edita and Lago Augusta are located. Also shown are palynological sites
discussed in the main text.

Figure 2. Stratigraphic column, radiocarbon dates and loss-on-ignition data from the Lago
Edita record. The labels on the right indicate the identity and stratigraphic span (dashed
horizontal lines) of each core segment.

Figure 3. Age model of the Lago Edita record, the blue zones represent the probability
distribution of the calibrated radiocarbon dates, the grey zone represents the calculated
confidence interval of the Bayesian age model.

Figure 4. Percentage pollen diagrams from the Lago Edita core.  The labels on the right
indicate the identity and stratigraphic span (dashed horizontal lines) of each pollen
assemblage zone. The black dots indicate presence of Drimys winteri pollen grains,
normally <2%.

Figure 5. Macroscopic charcoal record from the Lago Edita core and results of
CharAnalysis: blue line: background component, red line: locally defined threshold,
triangles: statistically significant charcoal peaks, magnitude: residual abundance that
supersedes the threshold.



Figure 6. Selected palynomorph abundance of the Lago Edita record shown in the time
scale domain. The red lines correspond to weighted running means of seven adjacent
samples with a triangular filter. The taxa shown in the left panel are characteristic of
humid environments currently found in sectors adjacent to the Pacific coast and/or the
Andean treeline in the study area. The taxon *Nothofagus dombeyi* type, which includes
multiple species with contrasting climatic tolerances, is also found in (relatively) humid
sectors east of the Andes. The herbs and shrubs shown in the right panel are either
cosmopolitan or present in the Patagonian Steppe and sectors located at or above the
Andean treeline in central-west Patagonia.

Figure 7. Comparison of selected tree pollen recorded in Lago Fácil, Lago Oprasa, Lago
Stibnite (Lumley and Switsur, 1993) and Lago Edita. The red line corresponds to a
weighted running mean in each record of seven adjacent samples with a triangular filter.
The lower panels show the curves from all sites expressed in a common percent scale
(Lago Fácil: purple line, Lago Oprasa: blue line, Lago Stibnite: black line, and Lago Edita:
red line).

Figure 8. Comparison of the percent sum of arboreal pollen (AP) in records from Lago
Edita, Lago Stibnite (Lumley and Switsur, 1993) and the spliced Canal de la Puntilla-
Huelmo time series (Moreno et al., 2015), as proxies for local rise in treeline driven by
deglacial warming. These data are compared against the δ Deuterium record from the
Antarctic Epica Dome Concordia (EDC) ice core (Stenni et al., 2010), and hydrologic
estimates from northwestern Patagonia. The latter consist of the percent abundance of
Magellanic Moorland species found in the spliced Canal de la Puntilla-Huelmo record
(Moreno et al., 2015), indicative of a hyperhumid regime, and the percent abundance of
the littoral macrophyte *Isoetes savatieri* from Lago Lepué (Pesce and Moreno, 2014),
indicative of low lake level (LL) during the earliest stages of T1 and the early Holocene
(9000-11,000 yr BP). The vertical dashed lines constrain the timing of the early Holocene
SWW minimum at 41°-43°S (9000-11,000 yr BP) (Fletcher and Moreno, 2011), a low-



precipitation phase during the early termination at 41°-43°S (16,800-17,800 yr BP)
associated with a southward shift of the SWW (Pesce and Moreno, 2014), the final LGM
advance of piedmont glacier lobes (17,800-19,300 yr BP) and the final portion of the Varas
interestade (19,300-21,000 yr BP) in the Chilean Lake District (Denton et al., 1999; Moreno
et al., 2015). The dashed green horizontal lines indicate the mean AP of each pollen record
prior to their increases during T1 (Lago Edita: 17%, Lago Stibnite:2%, spliced Canal de la
Puntilla-Huelmo: 31%). The ascending oblique arrow represents a northward shift of the
SWW, the descending arrow a southward shift of the SWW at the beginning of T1.




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





Table 1

| Laboratory code | Core | Material | Length (cm) | 14C yr BP±1σ | Median probability (cal yr BP) | 2σ range (cal BP) |
|---|---|---|---|---|---|---|
| UCIAMS-133501 | PC0902AT7 | Bulk | 660-661 | 8935±25 | 10,029 | 9794-10,177 |
| UCIAMS-133416 | PC0902AT8 | Bulk | 705-706 | 11,350±60 | 13,229 | 13,109-13,350 |
| UCIAMS-133417 | PC0902AT8 | Bulk | 757-758 | 13,740±70 | 16,863 | 16,684-17,055 |
| UCIAMS-133418 | PC0902AT8 | Bulk | 795-796 | 16,250±90 | 19,414 | 18,934-19,779 |
| CAMS-144454 | PC0902BT8 | Bulk | 795-796 | 16,020±50 | 19,164 | 18,922-19,408 |





Figure 1

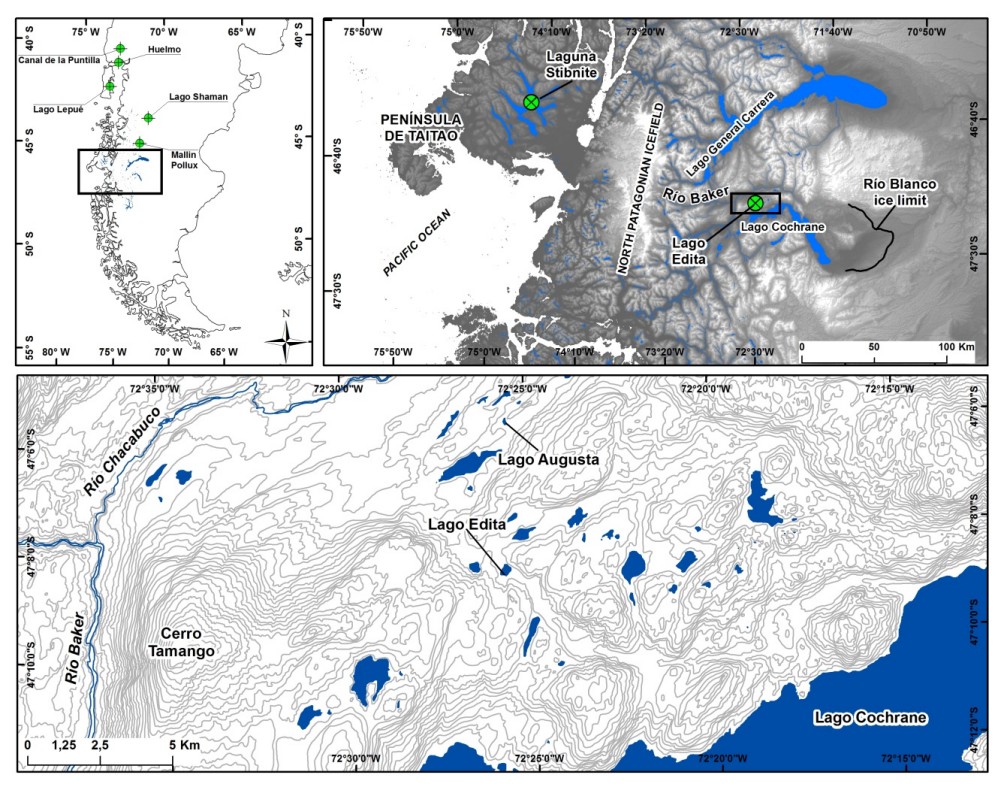







Figure 2




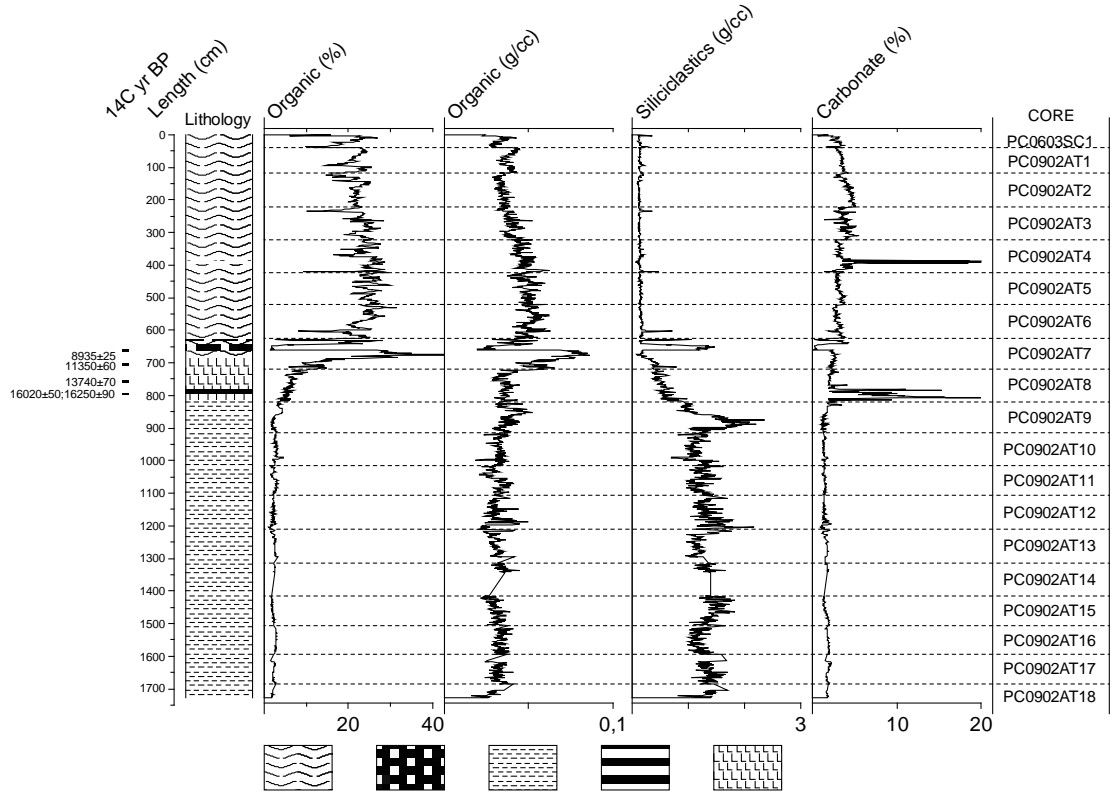









Figure 3

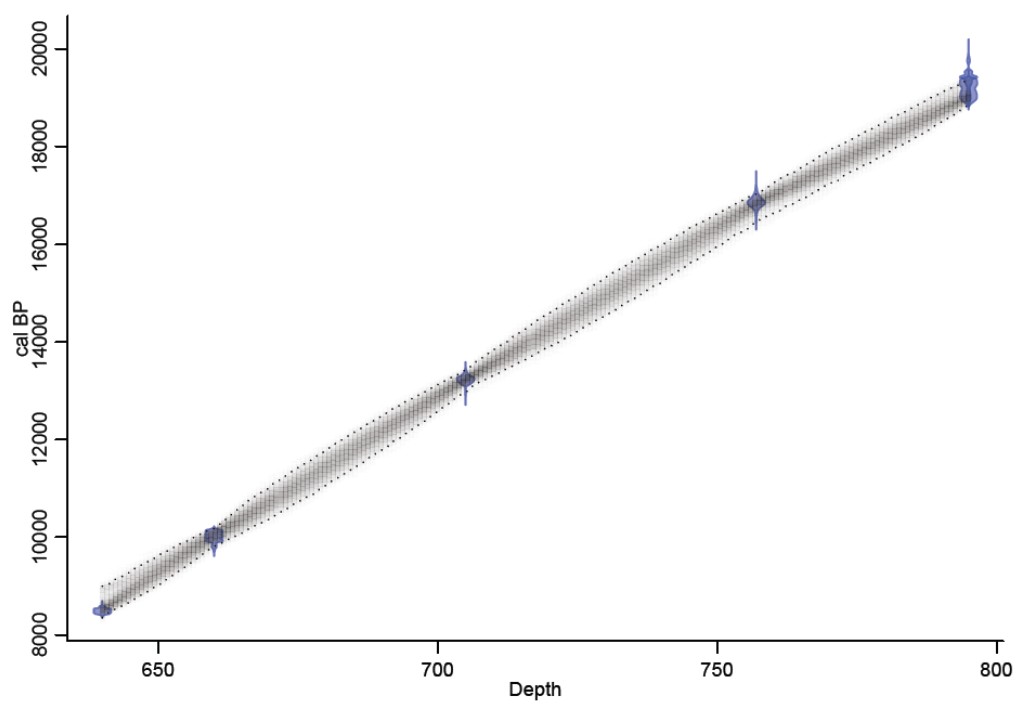







Figure 4

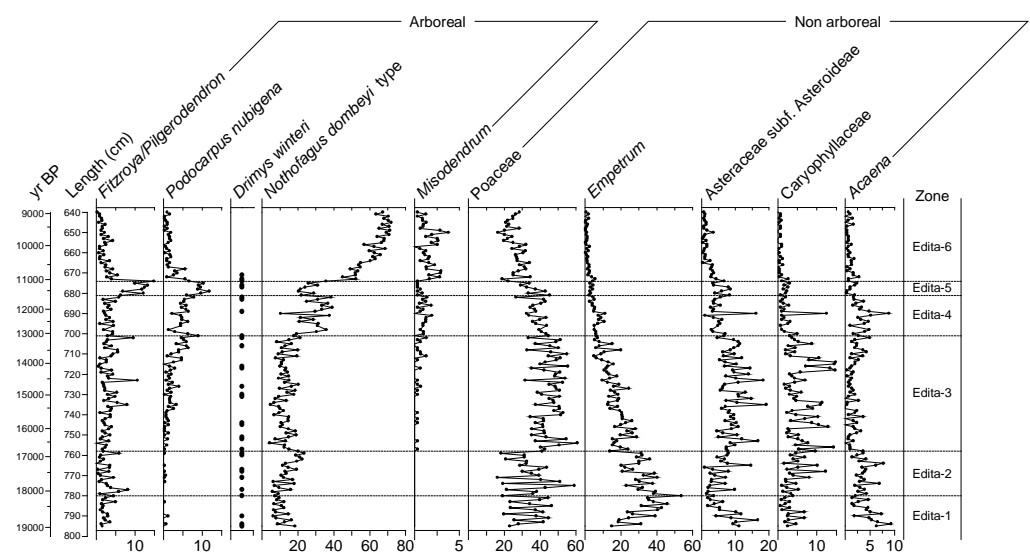


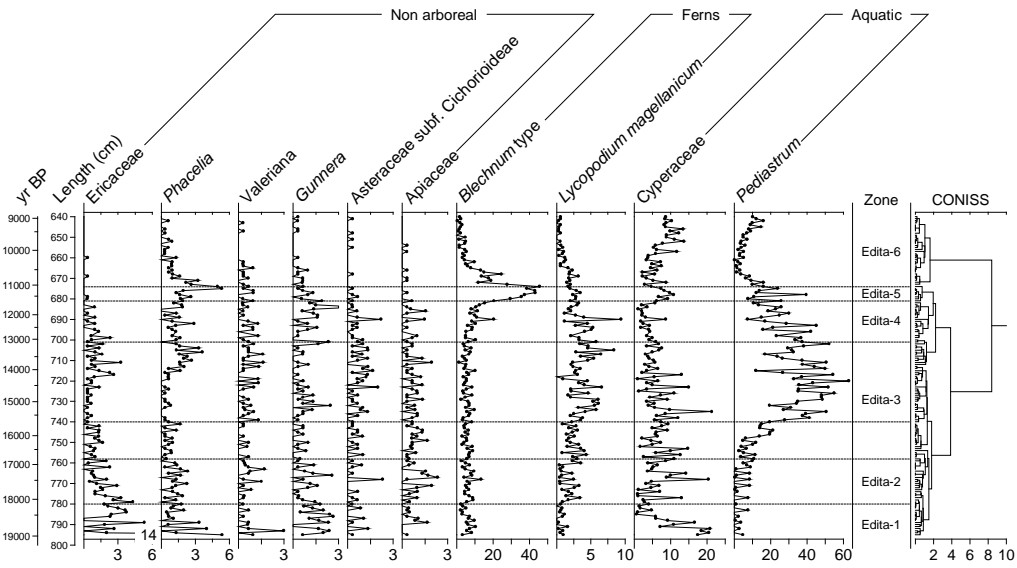




Figure 5

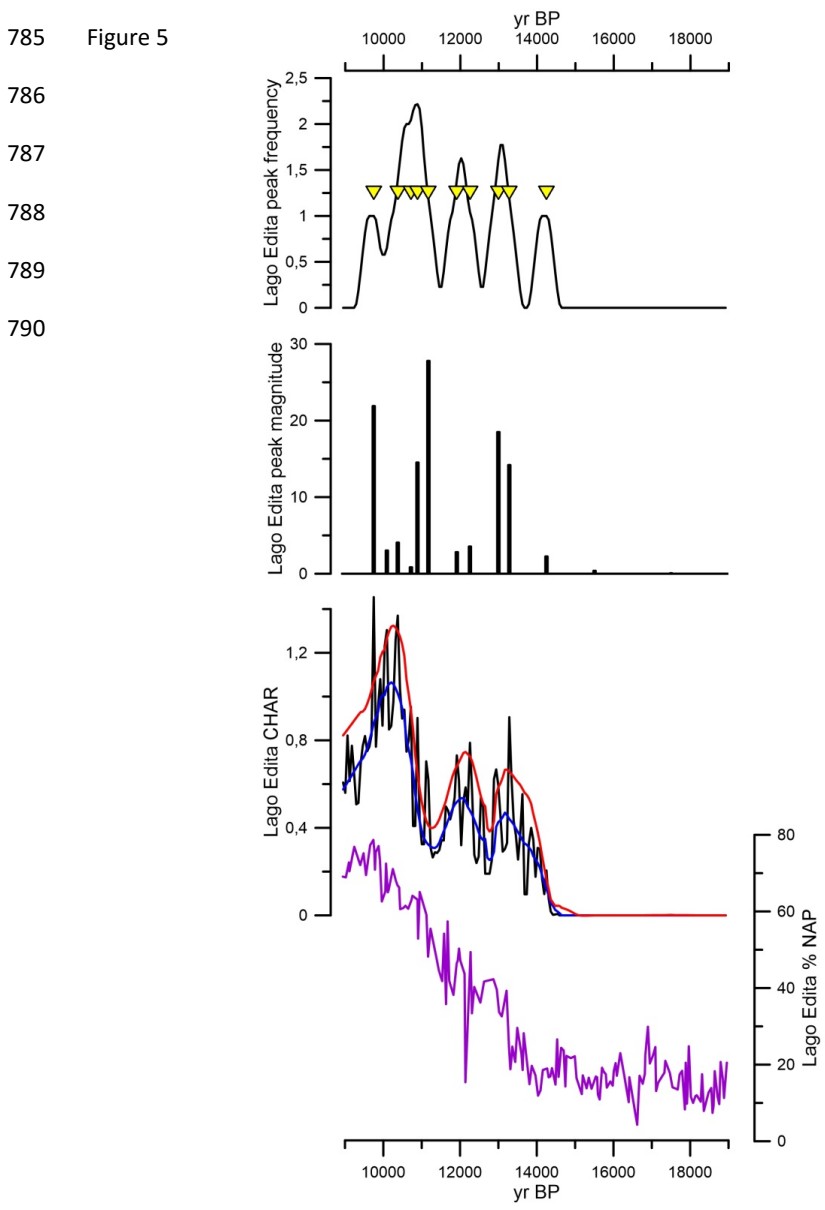

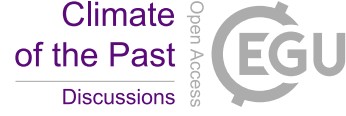



Figure 6

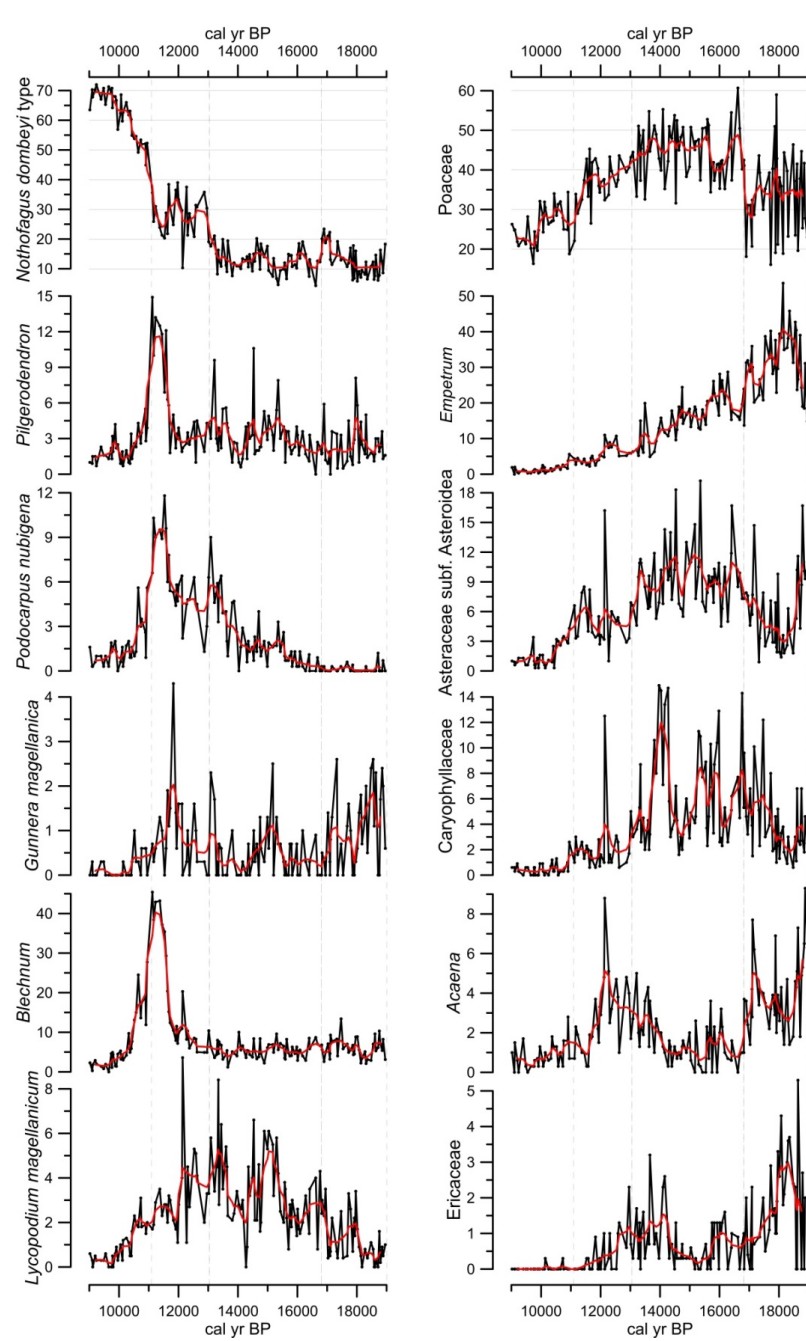





Figure 7

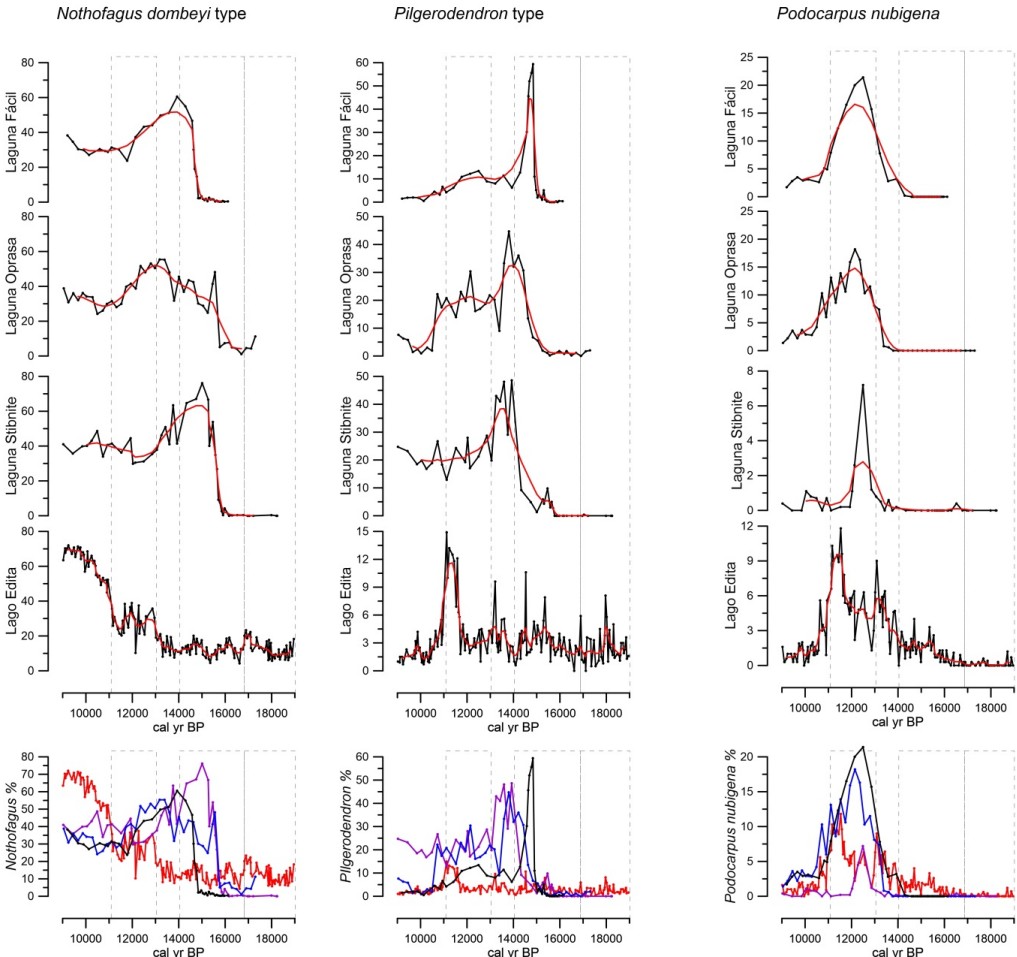






Figure 8


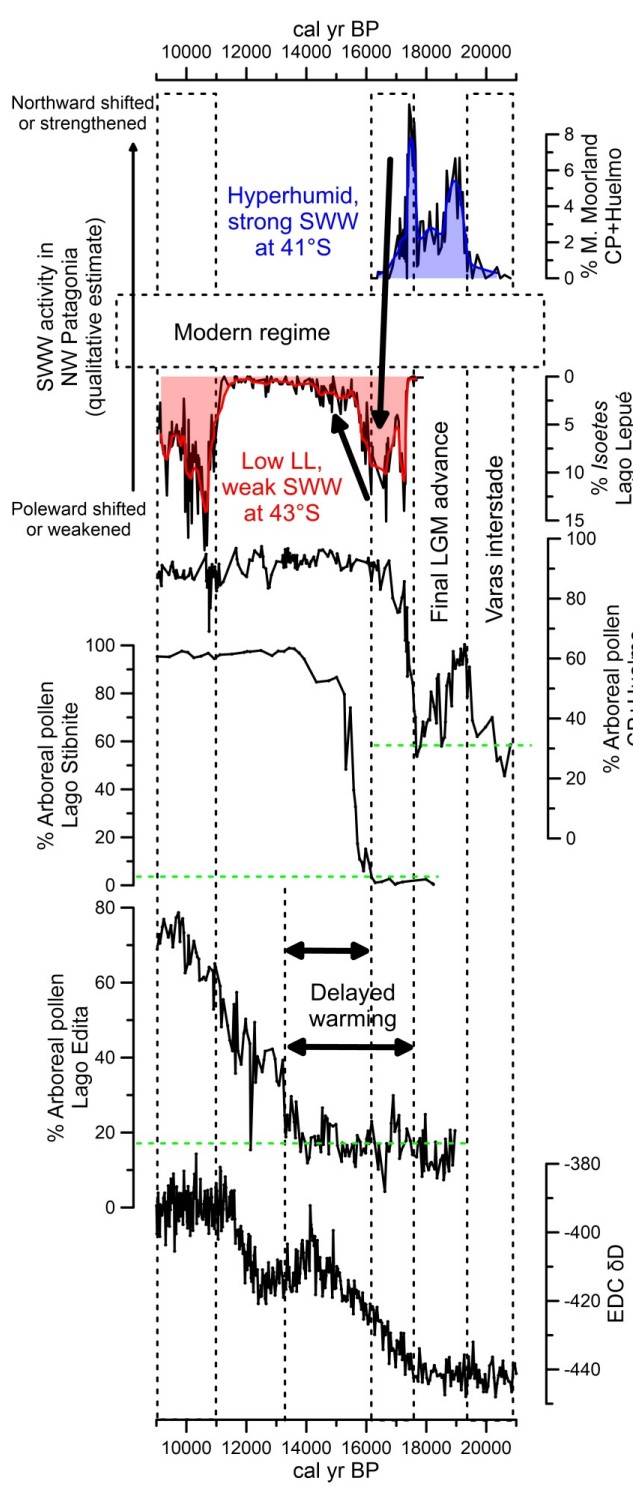