# Peer review of "THE LAST GLACIAL TERMINATION ON THE EASTERN FLANK OF THE CENTRAL PATAGONIAN ANDES (47°S)"

_Climate of the Past, 2016_

## Referee Comment (RC1) · Anonymous Referee #1 · 12 Nov 2016

Review of manuscript entitled "The last glacial termination on th eastern flank of the central Patagonian Andes (47 deg S) by William I. Henriquez et al.

This manuscript reports the results of a study into lake cores in the valle chacabuco in central Chilean Patagonia in an area formerly occupied by a major outlet glacier of the former Patagonian Ice Sheet east of the present day North Patagonian Icefield. The study reports palaeoecological changes in this small closed-basin lake, which appear to indicated deglaciation of ∼90 km had occurred in the valley by 19.4 ka, and that warming episodes occurred at 13.2 and 11 ka, with the latter being linked to the breakup of the north and south sectors of the former Patagonian ice sheet.

I am not a palaeoecologist and therefore, unfortunately, I am unqualified to comment on the data that is central to this manuscript. Instead, I focus my review on the overall

implications of the data as interpreted wrt the glacial history in this valley, and a few general comments. From my review of these aspects of the manuscript, I believe the work to be a valuable contribution of new data in a region that is generally lacking detailed palaeoclimate records. My recommendation is for publishing the manuscript (providing other reviewers view the data positively) following minor revision.

General comments:

- The figures are poor.

Figure 1 is the only figure that gives any sort of context to the study, but it does not include many of the place names discussed in the text, making it very difficult to follow some of the discussion in the manuscript. For example when describing other records of deglaciation in the valley you mention: Sierra Colorado, Lago Esmeralda, Cerro Oportus, Maria Elena moraine, Lago Columna/Posada moraine – but none of these are shown on Figure 1. When revising the figures, ensure all of the areas discussed are shown on the figures.

The manuscript would benefit from a photo showing evidence for the upper lake terrace (591 m), which is used to infer two glacial dams in the valley.

I also felt the manuscript would benefit from a conceptual model of deglaciation in the valley that includes not only the changing ice extent, but also the changing vegetation over the time period of interest. This would be a simple way to help convey the results of your study.

- The structure of the introduction is awkward The focus of the paper is on central Patagonia, but the opening paragraph discusses deglaciation in NW Patagonia and southernmost Patagonia (Cordillera Darwin). This is followed by a paragraph explaining the lack of data in central Patagonia, and the importance of gaining knowledge in the central sector, which is built on in the next paragraph on the importance of these data for model simulations of past climate. Then it jumps back to discussing data on the

deglaciation of central Patagonia (which in the previous paragraph was described as missing or lacking). Finally, the present study is discussed. The introduction therefore is a bit awkward, and needs to be streamlined so that you don't jump between different ideas.

Specific comments:

- L45: Pacific Ocean rather than coast - L53: refer to Figure 1 for location - L43-60: its not clear why these sites are significant to the present study – more context required to make it relevant. - L61: I don't understand why you don't discuss the data from central Patagonia at the same time up front (L79-93)? It seems you should discuss what's been done before discussing outstanding questions and the need for more work. I realise that this paragraph is focused on western Patagonia, but it is not clear why the discussion in central Patagonia has been partitioned into east and west. - L75: You have not yet defined SWW (you do so on line 90). - L79: radionuclides (one word) - L79-86: Be more specific regarding what studies reported what results – perhaps break this into two or three sentences for clarity. - L83: 29kyr for the final LGM advance? Be clear what you mean by this. Are you reporting the timing of the maximum advance (outermost of the Rio Blanco moraines)? Or are you wanting to mention the final moraine before the onset of significant deglaciation (in this case, I guess the 'final limit' of 17.4 ka in Hein 2010). As written, it is not clear whether the date provided (incidentally, I have no idea where this number of 29 kyr comes from – wait it looks like you are taking data from Boex from Sierra Colorado rather than the rio Blanco moraines? You need to be more clear here) is meant to indicate the onset of deglaciation, or not. - L85: an event that took place... - L85: reword sentence - perhaps 'caused the breakup of North and south icefields... or something like that. The current wording is a mouthful and difficult to follow. - L90: define SWW earlier. - L95: collected from Lago Edita - L123: explain what is meant by 'changed the continenetal divide' to help reader understand the significant changes that occurred at the time. - L408-4099: Be more specific about the moraine limits, their relative

order and why not just report the recalculated ages for clarity. Perhaps something like: "Cosmogenic radionuclide dating of three main Rio Blanco moraine limits by Hein et al. (2010) yielded cosmogenic 10Be exposure ages, recently recalculated by Kaplan et al. (2011), of 28.7, 25.1 and 21.1 ka from outermost to innermost moraine, respectively" - obviously including uncertainties, etc. Do the same for the glacial recession and stabilisation dates too. -L413-416: Note that the age reported was considered a minimum age for the Lago Columna/Posada moraines. The inference is that these moraines should be older than 17.6 but younger than 19.6 ka. However, both of those age constraints are from single exposure dates from boulders, and thus should be considered no more than indicative. - L418: The Lago Esmeralda dates are from radiocarbon or cosmo or both? - L423: See comment above – 17.6 is a minimum. - L423: radionuclide - L415-429: A figure would help to convey this information – at present, very difficult to follow and Figure 1 contains none of the placenames. - L444: not clear what interpretation it is supporting. - L448: also report calibrated ages more clearly. - L459: So is there another explanation for why the radiocarbon ages could be old? Did Villa-Martinez provide an explanation for this outlier? Is there another reason why these radiocarbon ages could be so old? -L463: You should explain the order of the moraine belts earlier on, so it makes sense, perhaps in a figure. L465: Ok, there is actually an erratic from the Lago Columna moraine with an age of 20.0 ka; this mean is a bit akward since it is not necessarily from the same landform as such. L470: There is an error here: the "final LGM limit" and the Lago Columna/Posada moraines as reported by Hein 2010 are only estimated at between 19.6 and 17.6 ka, based on 2 exposure dates. Within uncertainty, this still fits with your deglaciation age of 19.4 ka. With respect to the Boex dates for Elena moraines, these do appear younger than the radiocarbon ages. One thing should be noted in the manuscript is that all of the cosmo ages reported are minimum ages. This means that, for example, no erosion correction is applied. If the rock has eroded at all, the age would be greater. Likewise, if the rock is seasonally shielded by snow the age would get older. If the area has isostatically uplifted post deglaciation, the age would get older. If the rocks were deposited within

a proglacial lake (thus shielded by water for some time), the age would get older. The point I want to make is that the cosmo ages are reported as minimums, and you might want to discuss some of these points when trying to reconcile the radiocarbon ages you obtain in your core. - L522: this lake isn't really upstream (SW) - L537: What are the implications for this wrt the Menucos moraines in Lago Buenos Aires (Douglass et al., 2006; Kaplan et al 2004)?

Please also note the supplement to this comment:
http://www.clim-past-discuss.net/cp-2016-89/cp-2016-89-RC1-supplement.pdf

---

## Referee Comment (RC2) · Anonymous Referee #2 · 6 Dec 2016

The paper presents a new pollen record from a sediment core of Lago Edita located at 570 m elevation on the east-side of the Andes at latitude 47°S and to the east (typical lee side) of the present-day Northern Patagonian Ice Field. The lake is located between two former glacial valleys which have been also partly covered by glacier lobes and partly filled by proglacial lakes during the LGM and the following ice retreat. Besides paleoecological considerations the pollen record was also used to deduce the ice retreat of the Patagonian ice sheet after the LGM until the beginning of the Holocene (Termination 1) as well as shifting of the Southern Hemispheric Westerlies. The general scope is appropriate for the addressed journal. The introduction of the paper is partly focusing on the previous ice retreat reconstructions from the Cordillera Darwin (54°S), the investigated area between 45 to 48°S and the southern region of the Chilean Lake District. Afterwards the authors mention the modelling of the thermal and atmospheric impact by the Laurentia ice sheet, probably to suggest that this could be also a scenario for the area to the east of the Northern Patagonian Ice field. However, I am not convinced that this can be easily compared with the situation in the working area. Finally they address severalk questions they like to address with their record. However, some basic information and/or introduction how the palynology of such an area can be interpreted and what problems could appear with such interpretations are not introduced. How could be the past climate conditions with 1 to >6°C lower temperatures and with different unknown humidity deduced from the pollen record? Ok., such aspects could be also addressed in the discussion, but I am also missing most of the following aspects in the discussion: âŮŔ What is the possible size and also the altitudinal distribution of the pollen catchments of the investigated site? âŮŔ Does the pollen represent a mixture of one or both associated valleys and its plant vegetation at different elevations? I agree that a far distance transport of from the coastal zone is not likely, since existing records form this area are different. âŮŔ Are such pollen records able to recognize change in the timberline and therefor could give implication for temperature changes? âŮŔ How does the tree growth react with respect to changes in precipitation, evaporation and/or changes in the soil moisture? Such aspects and other climatic and ecological implications from the pollen/paleovegetation spectra have been partly discussed for Holocene scenarios but not as much for the past. What implication have 5 to 6°C lower temperature during the LGM for the evaporation and soil moisture and the amount of plant available water. âŮŔ How fast is the development of a plant succession near the timberline and how fast does the pollen community and the ecosystem react on climate changes. How did the timberline changed during Termination 1 and does the pollen record provide information concerning this question? However, the regional position and extend of the proglacial lake system and changes in the ice margin of the glacier lobes are not well illustrated in Figure 1 for different periods of the glacier retreat. The paper includes many discussion concerning shifting and intensity of the westerlies during T1 which are deduced from the hypothesis that humidity and/or precipitation have been clearly

westerly-linked. Garreaud et al. (2013) has calculated the present day relationship between precipitation and westerly strength. I could imagine that at the investigated site a R-value of around 0.4 describes the correlation between precipitation and westerly strength based of NCEP/NCAR data of the past 40 years. But is this also valid for T1? Beside this, the effect of lower temperatures as well as weaker or stronger wind has an import effect on the evaporation and thus on the humidity of the soils, in particular on the east side of the Andes. Only one record of Moreno from further north was taken as implication for the paleotemperature development. Siani et al (2013) MD07/3088 record from around 47°, and further north at 41° S the ODP 1233 record and the MD07/3128 record of from 53°S (both shown and compared in Caniupan et al. 2011) provide further SST's which indicate a very strong temperature (around 5°C) increase between 18.0 to 15.5 Kyrs. Afterwards the between 15 and 11 Kyrs the temperature increase may be slightly, more stepwise and less pronounced (2°C). Concerning the near-costal SST development the former records gives a consistent overall picture, but at the investigated site the tree pollen starts to increase strongly first after 13 kyrs. To explain such a comparatively late forest expansion (also later than a record of Lake Stibney from the western side of the Andes indicates) the authors try to explain this "delay in warming" (marked in Fig. 8) by a persisting very long delay in regional warming (which would have also affected the glacier retreat dynamics in this area). However, I cannot believe that there was a delay in warming of about 4500 years at latitude 47° and that there have been such a strong temperature depression between the Westside and East side of glaciated Andes. If this is a realistic scenario it should be quantitatively better justified. The westerly behavior is very complex. Recently, the SWW strength and related precipitation has a summer minimum at latitudes 47°S and northward, whereas from 50 to 55°S it has a summer maximum. How does such seasonal pattern affect the plant communities and the investigated site and what catchment have they sampled in an area with very strong local climate gradients? This concerns also the vegetation changes with altitude. The investigate lake sediment record is situated in an area of steep valleys. What does these pollen represent? The average plant community of the valley or above a certain elevation? What role plays the tree timberline or its changes? These questions are not well addressed. I am aware that the interpretation and discussion of these pollen is a very complex topic and obviously I am not a palynologist. But I would like to see the paleoclimate considerations better reviewed. There seems to be a lot of published work concerning the regional relationship between proglacial lake evolutions and ice retreat and glacier margins. How did this change the spatial distribution of plant growing areas and/or pollen catchments? There is something mentioned, but it is not well illustrated by maps. Finally, all the figures are very poorly and sluggish prepared and much information are missing. I made many comments in the text, but think that the coauthors have a lot of experience to review this seriously before resubmission. In conclusion, I suggest that the manuscript needs a major revision with respect to the above addressed topics. There are very experienced coauthors which published many good papers in international journals. I cannot believe that they have seriously revised and/or contributed to this paper. I made many comments in the text, but it will take several days to make a profound review of all things I would like to be improved. I think that the record data are important and should be published and I am willing to review this paper once more, but only after a profound revision of the coauthors.

Please also note the supplement to this comment:
http://www.clim-past-discuss.net/cp-2016-89/cp-2016-89-RC2-supplement.pdf

**Supplement:**

[revised manuscript text omitted]

Figure 1

[Figure]

[Figure]

Figure 2

[Figure]

[Figure]

[Figure]

[Figure]

Figure 3

[Figure]

[Figure]

[Figure]

Figure 4

[Figure]

[Figure]

[Figure]

Figure 5

[Figure]

[Figure]

[Figure]

Figure 6

[Figure]

[Figure]

Figure 7

[Figure]

[Figure]

[Figure]

Figure 8

[Figure]

[Figure]

---

## Author Comment (AC1) · 10 Jan 2017

We thank referee #1 for the constructive comments which helped improve the manuscript. About the general comments:

1. We improved Figure 1 as suggested. 2- Currently we do not thave a photo of the 591 ma.s.l. terrace fragment adjacent to Lago Edita. 3- The conceptual model, understood as a sequence of sketch maps showing ice margin, lake level, and vegetation distribution in the region at various time slices, requires information we currently do not have. Because we have not mapped geomorphic features we decided not to include such conceptual model. 4- We edited the introduction following referee #1's suggestions.

About the specific comments:

We incorporated all of the suggested changes in the text.

---

## Author Response (AR1)

Referee #1: "The figures are poor. Figure 1 is the only figure that gives any sort of context to the study, but it does not include many of the place names discussed in the text, making it very difficult to follow some of the discussion in the manuscript. For example when describing other records of deglaciation in the valley you mention: Sierra Colorado, Lago Esmeralda, Cerro Oportus, Maria Elena moraine, Lago Columna/Posada moraine – but none of these are shown on Figure 1. When revising the figures, ensure all of the areas discussed are shown on the figures. The manuscript would benefit from a photo showing evidence for the upper lake terrace (591 m), which is used to infer two glacial dams in the valley"

Response: We improved Figure 1 as suggested. Currently we do not have a photo of the 591 ma.s.l. terrace fragment adjacent to Lago Edita.

Referee #1: "I also felt the manuscript would benefit from a conceptual model of deglaciation in the valley that includes not only the changing ice extent, but also the changing vegetation over the time period of interest. This would be a simple way to help convey the results of your study"

Response: The conceptual model, understood as a sequence of sketch maps showing ice margin, lake level, and vegetation distribution in the region at various time slices, requires information we currently do not have. Because we have not mapped geomorphic features we decided not to include such conceptual model.

Referee #1: "The structure of the introduction is awkward. The focus of the paper is on central Patagonia, but the opening paragraph discusses deglaciation in NW Patagonia and southernmost Patagonia (Cordillera Darwin). This is followed by a paragraph explaining the lack of data in central Patagonia, and the importance of gaining knowledge in the central sector, which is built on in the next paragraph on the importance of these data for model simulations of past climate. Then it jumps back to discussing data on the deglaciation of central Patagonia (which in the previous paragraph was described as missing or lacking). Finally, the present study is discussed. The introduction therefore is a bit awkward, and needs to be streamlined so that you don't jump between different ideas"

Response: We edited the introduction following referee #1's suggestions

Referee #1 Specific comments

Response: We incorporated all of the suggested changes in the text.

Referee #1: "L522: this lake isn't really upstream (SW)"

Response: we modified the sentence as follows:

"According to Turner et al. (2005) the LCIL stabilized and deposited moraines in Lago Esmeralda, located ~10 km upstream *along the glacier flowline* and ~240 m lower in elevation than Lago Edita, between 13,600-12,800 yr BP"

Reviewer #2 stated: "The introduction of the paper is partly focusing on the previous ice retreat reconstructions from the Cordillera Darwin (54◦S), the investigated area between 45 to 48◦S and the southern region of the Chilean Lake District. Afterwards the authors mention the modelling of the thermal and atmospheric impact by the Laurentia ice sheet, probably to suggest that this could be also a scenario for the area to the east of the Northern Patagonian Ice field. However, I am not convinced that this can be easily compared with the situation in the working area"

Response: we do not intend to compare the thermal and atmospheric impact of the Laurentide ice sheet with the likely effect of the Patagonian ice sheet (PIS) along its eastern margin. We specifically stated that:

*"This aspect has remained largely unexplored in the PIS region, and might be a factor of importance for understanding the dynamics of the SWW and climatic/biogeographic heterogeneities through T1 at regional scale"*

Reviewer #2 stated: "How could be the past climate conditions with 1 to >6◦C lower temperatures and with different unknown humidity deduced from the pollen record?

Response: The basis of the palynological method is that indicator plant taxa from vegetation communities segregated along modern climate space enables reconstruction of past conditions revealed by fossil pollen records. Our record shows predominance of herbs and shrubs characteristic of modern alpine environments during the interval between 19,400-13,200 yr BP, accompanied by hygrophilous cold-resistant trees characteristic of the modern forests in the hyperhumid sector of coastal central Patagonia. We interpret this assemblage as indicative of cold and wet conditions over that interval. Arboreal increases after 13,200 yr BP suggest to us colonization of woodland and forest vegetation in the lowlands under warmer conditions. We envision that warming elicited a rise in the temperature-sensitive treeline, causing an increase in arboreal vegetation in the Río Chacabuco Valley.

Reviewer #2 stated: "Ok., such aspects could be also addressed in the discussion"

Response: Those aspects are described in the introduction, subsection study area. We decided to include additional information in the description of winter deciduous forests to further substantiate the point:

*"A study of the spatial and temporal variation in N. pumilio growth at treeline along its latitudinal range (35°40´S-55°S) in the Chilean Andes (Lara et al., 2005) showed that temperature has a spatially larger control on tree growth than precipitation, and that this influence is particularly significant in the temperate Andes (> 40°S). These results suggest that low temperatures are the main limiting factor for the occurrence of woodlands and forests at high elevations in the Andes, considering that precipitation increases with elevation at any given latitude (Lara et al., 2005). The modern treeline near Cochrane lies between 800-1180 m.a.s.l."*

Reviewer #2: "What is the possible size and also the altitudinal distribution of the pollen catchments of the investigated site?"

Response: we added information and a sentence to the final paragraph in the introduction section:

"In this study we report high-resolution pollen and macroscopic charcoal records from sediment cores we collected from Lago Edita (47°8'S, 72°25'W, ~570 m.a.s.l.), *a medium-sized closed-basin lake (radius ~250 m)* located in Valle Chacabuco, east of the central Patagonian Andes (Figure 1). *The relevant source area for pollen from lakes of this size is about 600-800 m from the lake's edge, according to numerical simulations using patchy vegetation landscapes (Sugita 1994)*"

Reviewer #2: "Does the pollen represent a mixture of one or both associated valleys and its plant vegetation at different elevations?"

Response: we added the following sentence to the beginning of the discussion section

*Given the size of Lago Edita (radius ~250 m), its pollen record is adequate to reflect local vegetation within 600-800 m from the lake's edge. An extra-local component is also present considering that species of the genus Nothofagus also produce large quantities of pollen grains susceptible to long-distance transport (Heusser, 1989). These attributes suggest that the Lago Edita fossil record might be a good sensor of vegetation located on the western end of Valle Chacabuco and the Lago Cochrane basin.*

Reviewer #2: "I agree that a far distance transport of from the coastal zone is not likely, since existing records form this area are different. Are such pollen records able to recognize change in the timberline and there for could give implication for temperature changes?"

Response: we concur; our manuscript is centered on the concept that a rise in treeline at the end of the last glaciation, driven by climate warming, led to the colonization and densification of arboreal vegetation in Valle Chacabuco.

Reviewer #2: "How does the tree growth react with respect to changes in precipitation, evaporation and/or changes in the soil moisture?"

Response: Our paper does not dwell on tree-growth patterns. We provide a succinct description of the regional vegetation composition and distribution, along with a reference to the Lara et al. (2005)'s study. Reviewer #2's question is peripheral to the main scope of our study, perhaps the pertinent ecophysiological literature might be more appropriate to address this question.

Reviewer #2: "What implication have 5 to 6∘C lower temperature during the LGM for the evaporation and soil moisture and the amount of plant available water?"

Response: this aspect has not been modelled in the study area. Simulation of these variables under different scenarios of temperature change and SWW strength will certainly shed light into this unexplored aspect.

Reviewer #2: "How fast is the development of a plant succession near the timberline and how fast does the pollen community and the ecosystem react on climate changes?"

Response: little detailed information is available for the Patagonian region during the last glacial termination. Terrestrial records from northwestern Patagonia indicate indistinguishable radiocarbon-dated chronologies for the response of the vegetation and glacial system at the onset of the last glacial termination. We added a brief reference to this rapid vegetation change in the first paragraph of the introduction:

"These data, along with the Canal de la Puntilla-Huelmo pollen record (~41°S) (Moreno et al., 2015) (Figure 1), indicate abandonment from the LGM margins in the lowlands at 17,800 yr BP, *abrupt arboreal expansion,* and accelerated retreat that exposed Andean cirques located above 800 m.a.s.l. within 1000 years or less in response to abrupt warming"

Reviewer #2: "How did the timberline changed during Termination 1 and does the pollen record provide information concerning this question?"

Response: our manuscript is centered on this subject.

Reviewer #2: "However, the regional position and extend of the proglacial lake system and changes in the ice margin of the glacier lobes are not well illustrated in Figure 1 for different periods of the glacier retreat"

Response: We added information in the new figure 1 based on published material. Some information, however, is yet unknown (varying extent of GLC through T1).

Reviewer #2: "The paper includes many discussion concerning shifting and intensity of the westerlies during T1 which are deduced from the hypothesis that humidity and/or precipitation have been clearly westerly-linked. Garreaud et al. (2013) has calculated the present day relationship between precipitation and westerly strength. I could imagine that at the investigated site a R-value of around 0.4 describes the correlation between precipitation and westerly strength based of NCEP/NCAR data of the past 40 years.
But is this also valid for T1?"

> Response: this aspect has not been modelled with the required detail to address this question in the study area. Downscaling of GCM simulations along a time-continuum through T1 will certainly shed light into this unexplored aspect.

Reviewer #2 stated: "Only one record of Moreno from further north was taken as implication for the paleotemperature development. Siani et al (2013) MD07/3088 record from around 47°, and further north at 41° S the ODP 1233 record and the MD07/3128 record of from 53°S (both shown and compared in Caniupan et al. 2011) provide further SST's which indicate a very strong temperature (around 5°C) increase between 18.0 to 15.5 Kyrs. Afterwards the between 15 and 11 Kyrs the temperature increase may be slightly, more stepwise and less pronounced (2°C)"

> Response: Apparently reviewer #2 wants us to reference SST changes during T1. We added the following sentence at the end of the first paragraph of the introduction:
>
> *"Sea surface temperature records from the SE Pacific (Caniupán et al., 2011) are consistent with these terrestrial records, however, their timing, structure, magnitude and rate of change may be overprinted by the vicinity of former ice margins and shifts in marine reservoir ages (Caniupán et al., 2011; Siani et al., 2013)"*

Reviewer #2: "I cannot believe that there was a delay in warming of about 4500 years at latitude 47° and that there have been such a strong temperature depression between the Westside and East side of glaciated Andes. If this is a realistic scenario it should be quantitatively better justified"

> Response: We have a different view on this subject and provide paleovegetation data to substantiate a thermal contrast across the Andes at latitude 47°S during T1. Quantitative estimates of temperature change in terrestrial environment east and west of the Andes are currently unavailable and, therefore, reviewer #2's expectations cannot be fulfilled with current knowledge.

Reviewer #2: "How does such seasonal (*precipitation*) pattern affect the plant communities and the investigated site and what catchment have they sampled in an area with very strong local climate gradients?"

> Response: our description of the regional patterns of vegetation composition and distribution addresses this point, along with the additions we made in the revised manuscript (see responses above).

Reviewer #2: "The investigate lake sediment record is situated in an area of steep valleys. What does these pollen represent?  The average plant community of the valley or above a certain elevation? What role plays the tree timberline or its changes?"

> Response: our revised manuscript and responses address these questions (see above).

Reviewer #2: "There seems to be a lot of published work concerning the regional relationship between proglacial lake evolutions and ice retreat and glacier margins. How did this change the spatial distribution of plant growing areas and/or pollen catchments? There is something mentioned, but it is not well illustrated by maps"

> Response: As stated in the original manuscript, plant colonization of the Valle Chacabuco could only occur once the area was ice free and devoid of proglacial lake influence at the elevations relevant for the Lago Augusta and Lago Edita areas. We did not include maps showing the distribution of the vegetation through T1 because the data from these two sites are insufficient to produce a spatially explicit view of vegetation change both east and west of the Andes.

Reviewer #2: "all the figures are very poorly and sluggish prepared and much information are missing"

> Response: we modified the figures that required improvements.

[revised manuscript text omitted]

Figure 2

[Figure]

Figure 3

[Figure]

Figure 4

[Figure]

[Figure]

Figure 5

[Figure]

Figure 6

[Figure]

Figure 7

[Figure]

Figure 8

[Figure]

---

## Referee Report (RR1)

Referee report

I am pleased to see that this manuscript presents a compelling and well-written vegetation reconstruction from a single site in Central Patagonia that covers most of the transition out of the last glaciations (T1). The data itself is high in detailed and quality, and the pollen analysis is excellent as usual for this group of investigators. I see clearly that the regional vegetation/climate relationships are satisfactorily explained in the *Study Area* section and supported by several atmospheric and palaeoclimate studies cited throughout the text. Thus, the climate interpretations of the pollen data, in particular the ones pertaining shifts of the Southern Westerly Winds (SWW), follow a reasonable logic, adding great scientist relevance to the interpretations.

Apart from minor mistakes regarding the figure captions, my main criticism of this manuscript is that the data presented in the result section do not provide with indisputable evidence to sustain the vegetation/climate trends inferred in the final part of the text, at least not in such emphatic manner. Alternative interpretations of the pollen trends, on the other hand, are heavily missed. In particularly, I do not see clear evidence for the allegedly continuous presence of several rainforest trees around the site during T1. The development of a parkland with scatter, rain-tolerant trees could have perfectly been the case in other areas of this region, and the authors may have unpublished or previously published data supporting the interpretations made from Lago Edita, but they should at any case be extremely careful about the broad and high-sounding climate assertions based on the pollen evidence presented in the manuscript. This is especially important since the past activity of the SWW is and highly relevant ongoing discussion among the palaeoclimate community today.

More specifically, in line 336 the authors mention that *Nothofagus* pollen is usually transported long distances and thus its deposition at Lago Edita does not necessarily translate into local presence of this taxon. However, further in line 352 about 15% of *Nothofagus* is interpreted as evidence for the local presence of beech trees around the site. I note that this interpretation is made despite that the specific *Nothofagus* mistletoe, *Misodendrum*, is completely absent before 16.8 ka, which to me highlights that fact that the extra-local component of *Nothofagus* is maximized. When taking into consideration that *Misodendrum* is not present in regular abundance before 14 ka, there isn't in my opinion robust evidence for sustain the presence of *Nothofagus* trees around the site before that time. This is not considered or discussed at all in the manuscript and it should be. As a result of this, the claimed presence of low density, scattered hygrophilous trees around the site between 19-14 ka relies uniquely on the discontinuous 3-5% of *Fitzroya/Pilgerodendron*, 0-3% of *P. nubigena* and 0-1% of *Drimys*. This is in my opinion a weak palynological signal to argue in favor of local hygrophilous vegetation. Actually, a look at the pollen diagram reveals the presence of several herbs listed as member of the "High Andean Desert" (i.e. Apiaceae, *Gunnera* and *Valeriana*) in percentages that are, in sum, well above the trees uses as wet indicators. Hence, a climate interpretation that includes a dry phase between 19-13 ka could equally be drawn. I strongly recommend discussing this in the manuscript.

In summary, the presence of a parkland of hygrophilous trees on this region during T1 could certainly be a possibility when considering other published or unpublished records, but my point is that the evidence from the single site presented in this manuscript is not concluding enough to propose a sequence of vegetation change valid for the whole eastern margin of the Patagonia Ice Sheet. Furthermore, a vegetation interpretation based on less that 5% of the total pollen is used to infer the dynamic of a hemispheric-scale atmospheric system such as the SWW, which is in my opinion going too far away with the data.

A similar high-sounding climate interpretation is made from the pollen trends observed after 13 ka, when the percentages of hygrophilous trees are heavily reduced and *Nothofagus* increases rapidly. At first, these changes can be interpreted as a drop in moisture. Yet, how can we know that the decrease in the hygrophilous trees is not an artifact of the relative increment of *Nothofagus* if not pollen influx data is supplied? I note that the abundance of the hygrophilous trees after 11 ka is actually not lower than the period 19-13 ka, with the latter being interpreted as a relative wet period. This seems to be contradictory. There is more. According to the authors, a trend towards decreasing precipitation is also suggested by rise in CHAR. Yet, could this rise result from the densification of the tree coverage and associated fuel continuity rather than exclusively due to a climate forcing? This possibility is only mentioned but not taken into account in the interpretations. Additionally, to my understanding the time for the rapid CHAR increases is within the interval of the first *H. Sapiens* colonization of South America. Perhaps the CHAR rise was associated with the appearance and intensification of human-related ignition events summed to a more continuous distribution of fuel. Yet again, none of these arguments are discussed in proper depth in the main text.

In summary, my general impression is that some of the interpretations should be toned down and that alternative scenarios should also be considered and discussed in more detailed. It seems to me that the authors are trying to push the data to match the regional climate trends inferred from previously published pollen profiles, dismissing any alternative interpretation of the data. Beyond these important caveats, this is a persuasive and well-written manuscript.

Minor considerations

The authors rule out the long distance transport of *Nothofagus* since its presence has not been documented in western sites, but then again only two sites westward from the Andes are mentioned in the text attesting for the extremely low density of pollen profiles in a vast geographic region extending for several hundred kilometers across the pacific coast. Thus, in my opinion a western source from the *Nothofagus* pollen grains found at Lago Edita cannot be completely dismissed.

Please note that there are several mistakes in the Figure captions and their correspondence citations in the main text. Here I list a few of them, but I would recommend verifying that all the terminology used in the captions match the one used in the manuscript.

1. The figure captions do not state what "NAP" means in Figure 5. I am lean toward "Non-Arboreal Pollen", although this figure is cited in Line 322 to mention a correspondence between CHAR and % of *Nothofagus*. This is very confusing, please clarify.

2. In figure 4, the zones in the upper diagram are different to the zones in the lower diagram. There is a gap between zone Edita-2 and zone Edita-3 in the lower diagram.

3. Figure 2 is cited in Line 449 to mention the observation of massive pebbly layer as increases in the "Inorganic density data". Yet, there is no such as thing as "Inorganic density" in Figure 2 so that the reader is impeded to check the deposition of the pebble layers.

4. A look at Figure 2 reveals that organic sedimentation actually starts at about 700 cm or about 13 ka, and not at 19 ka as stated in Line 480.

5. There is no precipitation data for the High Andean Desert (Line 179); whereas all other vegetation zones have rainfall ranges.

6. In Line 204 it is mentioned that Lycopodium tablets were added to calculate pollen concentration and accumulation rate, but none of this data is provided.

7. The timing for the culmination of glacial advances in New Zealand is commented in Line 575 without any reference. Please add the corresponding citation.

---

## Editor Decision (ED1)

[revised manuscript text omitted]

Figure 2

[Figure]

Figure 3

[Figure]

Figure 4

[Figure]

[Figure]

Figure 5

[Figure]

Figure 6

[Figure]

Figure 7

[Figure]

Figure 8

[Figure]

---

## Author Response (AR2)

Dear Helen Bostock,

Editor SHAPE special issue

We have incorporated changes in the manuscript following the reviewer's suggestions and criticisms, namely:

Reviewer #1:

We were unable to incorporate the external errors on the 10Be moraine ages because not all of the original papers include that information.

We modified the sentence in the following manner:

stabilization and deposition of the Lago Columna and Lago Posada moraines **before** 17,600±900 yr BP

Reviewer #2:

"the data presented in the result section do not provide with indisputable evidence to sustain the vegetation/climate trends inferred in the final part of the text, at least not in such emphatic manner"… "there isn't in my opinion robust evidence for sustain the presence of *Nothofagus* trees around the site before that time. This is not considered or discussed at all in the
manuscript and it should be"

Response:
We included a sentence to address this issue:

"We note that the *Nothofagus* parkland on the western end of Valle Chacabuco and the Lago Cochrane basin must have approached the vicinity of Lago Edita at 16,800 yr BP, judging from the appearance of *Misodendrum* at that age (Figures 4, 6) under relatively constant mean *Nothofagus* abundances"

Reviewer #2:

"A similar high-sounding climate interpretation is made from the pollen trends observed after 13 ka, when the percentages of hygrophilous trees are heavily reduced and *Nothofagus* increases rapidly. At first, these changes can be interpreted as a drop in moisture. Yet, how can we know that the decrease in the hygrophilous trees is not an artifact of the relative increment of *Nothofagus* if not pollen influx data is supplied? I note that the abundance of the hygrophilous trees after 11 ka is actually not lower than the period 19-13 ka, with the latter being interpreted as a relative wet period. This seems to be contradictory"

Response:
We thank reviewer #2 for highlighting this problem. We realized it was necessary to include a paragraph and supplement several sentences to provide the required context:

"The conifer Podocarpus nubigena remained in low abundance (<2%) prior to ~14,500 yr BP in the Lago Edita record, increased between 14,500-13,000 yr BP, experienced a variable decline between 13,000-11,800 yr BP, reached a maximum between 11,800-11,200 yr BP and declined between 11,200-10,200 yr BP (Figures 4, 6). This cold-resistant hygrophilous tree is commonly found in temperate evergreen rainforests along the Pacific coast of central Patagonia and is currently absent from the eastern Andean foothills at the same latitude. Its presence and variations in the Lago Edita record suggest an increase in precipitation relative to the pre-14,500 yr BP conditions, with millennial-scale variations starting at ~13,000 yr BP. The variable decline in P. nubigena at 13,000 yr BP coincided with an increase in Nothofagus that led to a variable plateau of ~30% between 13,000-11,200 yr BP we will discuss in the following paragraphs"

"Declines and virtual disappearance of the cold-resistant hygrophilous trees *Fitzroya/Pilgerodendron*, *Podocarpus nubigena* along with the herbs *Gunnera magellanica* and *Lycopodium magellanicum* took place at ~11,000 yr BP in the Lago Edita record (Figures 4, 6), in response to a sudden decline in precipitation **relative to the ~14,500-11,000 yr BP interval"**

"We interpret these increases as treeline-rise episodes driven by warming pulses coupled with a decline in SWW strength at 47°S **(relative to the ~14,500-11,000 yr BP interval)**, as suggested by the disappearance of cold-resistant hygrophilous trees and herbs at ~11,000 yr BP"

Reviewer #2:

According to the authors, a trend towards decreasing precipitation is also suggested by rise in CHAR. Yet, could this rise result from the densification of the tree coverage and associated fuel continuity rather than exclusively due to a climate forcing?

We added the following sentence to address this point:

[revised manuscript text omitted]

---

## Author Response (AR3)

Tuesday, May 30, 2017

Dear Helen Bostock,

Editor SHAPE special issue

Thank you for your suggestions and corrections, all of which we incorporated in the new version. In the following pages you will be able to track those changes.

Sincerely,

Patricio Moreno

[revised manuscript text omitted]

Figure 2

[Figure]

Figure 3

[Figure]

Figure 4

[Figure]

Figure 5

[Figure]

Figure 6

[Figure]

Figure 7

[Figure]

Figure 8

[Figure]

---

## Author Response (AR4)

Dear Helen Bostock,

Editor SHAPE special issue

Thank you for your additional suggestions and corrections, which we incorporated in the R4 version. In the following pages you will be able to track those changes.

Sincerely,

Patricio Moreno

[revised manuscript text omitted]

We thank the editor and three anonymous reviewers for their constructive comments to early versions of this paper.

[revised manuscript text omitted]

Figure 2

[Figure]

Figure 3

[Figure]

Figure 4

[Figure]

Figure 5

[Figure]

Figure 6

[Figure]

Figure 7

[Figure]

Figure 8

[Figure]